# Biomarkers of response to neoadjuvant palbociclib plus anastrozole in endocrine-resistant estrogen receptor-positive/HER2-negative breast cancer: a phase 2 trial

Tim Kong [1,2,3,4], Alex Mabry [1,2], Maureen Highkin[1], Anthony Z. Wang[5], Jeremy Hoog[1], Zhanfang Guo[1], Adrian Gonzales-Gonzales[1], Shana Thomas[1], Yingduo Song [1], Feng Gao[6], Mateusz Opyrchal [1], Lindsay Peterson[1], Foluso Ademuyiwa [1], Julie Margenthaler[7], Rebecca Aft[7], Katherine Glover-Collins[7], Leslie Nehring[1], Yu Tao[6], Souzan Sanati[8], Ian S. Hagemann [8], Fouad Boulos[8], Matthew Holt[9], Li Ding [1], Wenge Zhu [10], Stephen T. Oh [3,8,11], Jianxin Wang [12], Agnieszka K. Witkiewicz[12,13], Erik S. Knudsen [12], Ron Bose [1], Jason D. Weber [1], Matthew Goetz[14], Donald Northfelt[15], Jingqin Luo [6,16] ✉ & Cynthia X. Ma [1,16] ✉

CDK4/6 inhibitors (CDK4/6i) improve outcomes for estrogen receptor (ER) positive/HER2-negative breast cancers (BCs), yet intrinsic and acquired resistance exist. Here, we evaluated anastrozole in combination with palbociclib (ANA/PAL) in the NeoPalAna Endocrine-Resistant cohort (NCT01723774). Thirty-four patients with clinical stage II/III ER + /HER2− BCs resistant to standard neoadjuvant endocrine therapy (on-treatment Ki67 > 10%) received neoadjuvant ANA/PAL, with serial biopsies analyzed. The primary endpoint, complete cell cycle arrest (CCCA; $Ki67_{C1D15} \leq 2.7\%$ at cycle 1, day 15), was achieved in 57.6% of patients (95%CI: 39.2–74.5%). Resistance to ANA/PAL ($Ki67_{C1D15} > 10\%$) was associated with higher pre-treatment tumor grade, Ki67, and specific PAM50 subtypes. Resistant tumors demonstrated reduced ER signaling and upregulation of cell cycle, mTOR, interferon, JAK/STAT, and immune checkpoints. Additionally, a 33-gene signature that predicted neoadjuvant Ki67 response to ANA/PAL was prognostic in a metastatic validation cohort. These findings underscore dysregulated oncogenic pathways as potential resistance mechanisms and biomarkers of response to CDK4/6i.

A major challenge in treating ER+/HER2− breast cancer (BC) is endocrine resistance[1]. The discovery of cyclin D/CDK4/6-Rb as the major effector of ER signaling and endocrine resistance mechanisms[2–6] led to the development of CDK4/6 inhibitors (CDK4/6is)[7]. These agents block Rb phosphorylation, which hinders E2F transcription factors from activating the expression of genes required for G1 to S phase transition[8]. Three CDK4/6is are FDA approved and have revolutionized the treatment for ER + BC[9,10]. However, despite their impressive anti-tumor activity, intrinsic and acquired resistance exist. There is a critical unmet need to understand the underlying mechanisms of resistance and developing response biomarkers[9,10].

The NeoPalAna trial (NCT01723774) is a multi-cohort single-arm phase 2 neoadjuvant study of the CDK4/6i palbociclib (PAL) plus anastrozole (ANA) in clinical stage II/III ER+/HER2− BC, aimed at assessing the antiproliferative effect of PAL in primary BC and discovering response biomarkers. We have previously reported the results of the initial cohorts (NeoPalAna Initial), including the *PIK3CA* mutated and the *PIK3CA* wild-type cohorts[11]. We observed a potent antiproliferative effect of PAL across a broad range of clinicopathologic and mutation profiles, including BCs resistant to ANA monotherapy. Few patients (6 of 50 patients) did not achieve CCCA (Ki67$_{C1D15}$ ≤ 2.7%). Analysis of these resistant cases suggested an association with high-risk PAM50 types and persistent E2F-target gene expression[11].

To further investigate PAL resistance mechanisms in endocrine therapy (ET)-resistant BC, we opened the Endocrine-Resistant (ET-Resistant; ET-R) cohort that enrolled patients with on-treatment Ki67 > 10% after ≥4 weeks of standard-of-care neoadjuvant ET (NET). The primary endpoint was CCCA at C1D15. Correlative endpoints included genomic and proteomic analyses of tumor tissue in relation to response categorized by Ki67$_{C1D15}$ ≤ 10% (ANA/PAL-sensitive; Sensitive) or >10% (ANA/PAL-resistant; Resistant).

## Results

### Patient characteristics

Between August 2016 and March 2021, 34 patients (17 pre- and 17 post-menopausal), with a median age of 52.8 years (range 30.4–77.6), and clinical stage II-III ER+/HER2− BC were enrolled. One patient lacked Ki67$_{C1D15}$ due to insufficient tumor material, leaving 33 patients evaluable for the primary endpoint (Fig. 1). Patient and tumor characteristics are detailed in Table 1. Notably, all but one patient had either Grade 2 (*n* = 16) or Grade 3 (*n* = 16) BC. Median Ki67 levels were 28.8% (range 7.5%–94.3%) at diagnosis (baseline; BL) and 30% (range 12%–96.4%) on NET prior to enrollment (C1D1). The majority had ER-rich BC with Allred scores of 8 (*n* = 26) and 7 (*n* = 3). All patients received standard-of-care neoadjuvant ANA (with goserelin if pre-menopausal) for a median duration of 6 weeks (range 4–12 weeks) prior to enrollment.

### Safety and clinical responses

All 34 patients were evaluable for treatment-related adverse events (TRAE) (Supplementary Table 1). The only G3 and above TRAEs were neutropenia (38% G3, 3% G4) and leukopenia (9% G3). Among the 34 patients enrolled, 13 patients went off ANA/PAL early due to Ki67 at C1D15 over 10% per protocol (*n* = 11) or intolerable AE (*n* = 2), The clinical response rate was 85.7% (18/21, 95% CI: 63.67–96.95%) in the 21 patients who completed 5 cycles of ANA/PAL (Table 2) and 52.5% (18/34) in the intent-to-treatment population.

### Primary endpoint and Ki67 response

Among the 33 evaluable patients, the C1D15 CCCA rate was 57.6% (19/33; 95% CI: 39.2–74.5%), which met the primary endpoint. Three of the 14 patients without CCCA had Ki67$_{C1D15}$ < 10%. Therefore, 22 (67%) were classified as ANA/PAL-sensitive (Sensitive) and 11 (33%) ANA/PAL-resistant (Resistant) using the Ki67$_{C1D15}$ 10% cut-point. Resistant tumors were more likely cT3 (*p* = 0.01), G3 (*p* = 0.018), and had higher Ki67 (*p* = 0.015) pre-treatment (Table 1). Race was associated with response (*p* = 0.036). However, caution should be applied in interpreting this result due to the small sample sizes: American Indian (*n* = 2), Black (*n* = 3), and Asian (*n* = 3), compared to the majority of White (*n* = 25).

### Baseline tumor mutation profile and copy number variations (CNV)

Thirteen patients had sufficient archival diagnostic biopsies for WES. To maximize sample size, we used C1D1 biopsies for 5 patients and C1D15 biopsies for 9 patients. The WES cohort included 8 (30%) Resistant, 18 (67%) Sensitive, and 1 with unknown Ki67$_{C1D15}$, similar to the overall cohort. Given the small sample size, we conservatively focused the analysis on the 83 genes most relevant for BC[12]. Consistent with the ET-Resistance phenotype, this cohort was enriched with genomic alterations implicated in endocrine resistance (Supplementary Fig. 1a)[13–16]. The top 4 frequently mutated genes were *TP53* (37%), *PIK3CA* (33%), *PTEN* (11%), and *NF1* (11%; Supplementary Data 1). We then compared the mutation frequencies of *TP53* and *PIK3CA* with the NeoPalAna Initial cohort (19 ET-Sensitive, 10 ET-Resistant)[11], Z1031_POL (48 ET-Sensitive, 28 ET-Resistant cases)[13], and the TCGA ER+/HER2− BCs (*n* = 396)[17]. *TP53* mutation frequency was significantly higher in the NeoPalAna ET-R cohort than that of the ET-Sensitive cohort of Z1031_POL (37% vs 8%, Fisher FDR *p* = 0.00845) (Supplementary Fig. 1b, c), as expected[13].

We then examined *TP53* and *PIK3CA* mutation frequencies in Sensitive and Resistant cases within the NeoPalAna ET-R cohort,

**NeoPalAna Endocrine-Resistant (ET-R) Cohort (NCT01723774)**

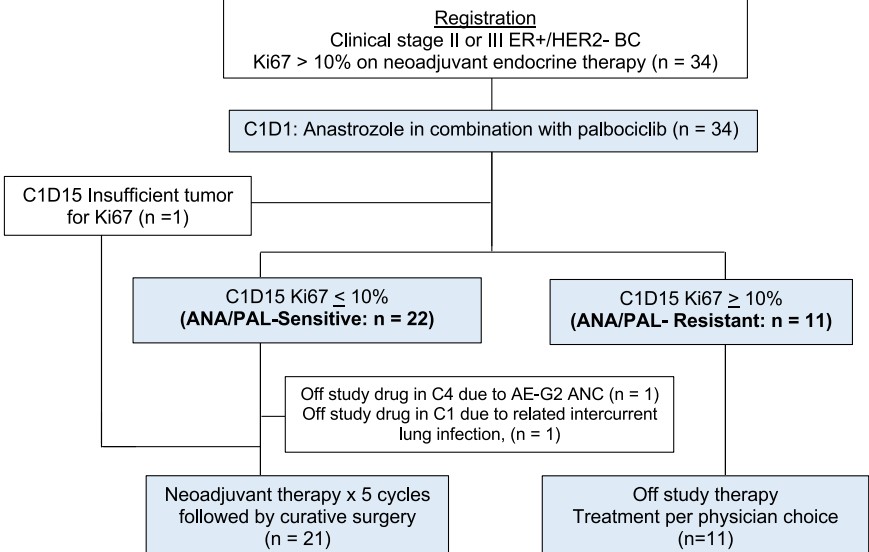

**Fig. 1 | CONSORT diagram of NeoPalAna Endocrine-Resistant cohort.**

**Table 1 | Clinical and tumor characteristics**

| Variable | Level | Overall (N = 33) | ANA/PAL-resistant (N = 11) | ANA/PAL-sensitive (N = 22) | P-Value |
|---|---|---|---|---|---|
| Age (median, SD) | | 52.8 (11.20) | 49.6 (10.47) | 54.5 (11.4) | 0.237 |
| Race | American Indian or Alaska Native | 2 (6.06%) | 1 (9.09%) | 1 (4.55%) | **0.036** |
| | Black or African American | 3 (9.09%) | 1 (9.09%) | 2 (9.09%) | |
| | Other–Asian | 3 (9.09%) | 3 (27.27%) | 0 (0.00%) | |
| | White or Caucasian | 25 (75.76%) | 6 (54.55%) | 19 (86.36%) | |
| Ethnicity | Hispanic or Latino | 6 (18.18%) | 1 (9.09%) | 5 (22.73%) | 0.637 |
| | Not Hispanic or Latino | 27 (81.82%) | 10 (90.91%) | 17 (77.27%) | |
| Postmenopausal | No | 16 (48.48%) | 7 (63.64%) | 9 (40.91%) | 0.218 |
| | Yes | 17 (51.52%) | 4 (36.36%) | 13 (59.09%) | |
| Stage | IB | 1 (3.13%) | 0 (0.00%) | 1 (4.76%) | 0.331 |
| | IIA | 12 (37.50%) | 3 (27.27%) | 9 (42.86%) | |
| | IIB | 13 (40.63%) | 4 (36.36%) | 9 (42.86%) | |
| | IIIA | 6 (18.75%) | 4 (36.36%) | 2 (9.52%) | |
| T stage | T1c | 1 (3.13%) | 0 (0.00%) | 1 (4.76%) | **0.010** |
| | T2 | 23 (71.88%) | 5 (45.45%) | 18 (85.71%) | |
| | T3 | 8 (25.00%) | 6 (54.55%) | 2 (9.52%) | |
| N stage | N0 | 10 (31.25%) | 5 (45.45%) | 5 (23.81%) | 0.513 |
| | N1 | 21 (65.63%) | 6 (54.55%) | 15 (71.43%) | |
| | NX | 1 (3.13%) | 0 (0.00%) | 1 (4.76%) | |
| Histology | Ductal | 32 (96.97%) | 11 (100.00%) | 21 (95.45%) | 1.000 |
| | Lobular | 1 (3.03%) | 0 (0.00%) | 1 (4.55%) | |
| Grade | I | 1 (3.03%) | 0 (0.00%) | 1 (4.55%) | **0.018** |
| | II | 16 (48.48%) | 2 (18.18%) | 14 (63.64%) | |
| | III | 16 (48.48%) | 9 (81.82%) | 7 (31.82%) | |
| PgR staus | Negative | 4 (12.12%) | 3 (27.27%) | 1 (4.55%) | 0.097 |
| | Positive | 29 (87.88%) | 8 (72.73%) | 21 (95.45%) | |
| ER allred score | 8 | n = 26 | n = 7 | n = 19 | 0.121 |
| | 7 | n = 3 | n = 1 | n = 2 | |
| | 6 | n = 1 | n = 1 | n = 0 | |
| | 5 | n = 1 | n = 1 | n = 0 | |
| | 4 | n = 1 | n = 0 | n = 1 | |
| | 3 | n = 1 | n = 1 | n = 0 | |
| C0D1 Ki67, median (range) | | 28.8% (7.5–94.3%) | 48.1% (27.0–94.3%) | 24.2% (7.5–55.8%) | **0.015** |
| C1D1 Ki67, median (range) | | 30% (12–96.4%) | 50% (17–96.4%) | 26.2% (12–89.8%) | 0.076 |
| C1D15 Ki67, median (range) | | 1.3% (<1–97.9%) | 36.3% (10.4–97.9%) | 1% (<1–2.5%) | **<0.001** |
| PAM50 subtype | Basal | 3 (13.64%) | 3 (37.50%) | 0 (0.00%) | **<0.001** |
| | HER2E | 2 (9.09%) | 2 (25.00%) | 0 (0.00%) | |
| | LumA | 9 (40.91%) | 0 (0.00%) | 9 (64.29%) | |
| | LumB | 8 (36.36%) | 3 (37.50%) | 5 (35.71%) | |

Bolded P-values indicate statistical significance (P < 0.05).

**Table 2 | Clinical response**

| Response category | N (%) N = 21 |
|---|---|
| Complete response | 3 (14.3%) |
| Partial response | 15 (71.4%) |
| Stable disease | 3 (14.3%) |
| Progressive disease | 0 |
| Response rate: 85.7% (95% CI: 63.67–96.95%) | |

(Supplementary Fig. 2a). CNVs of cell cycle mediators and commonly altered genes showed little difference between Sensitive and Resistant baseline tumors. However, significant CNVs in pathways including DNA repair and PI3K/AKT/mTOR signaling were more frequent in Resistant tumors (Supplementary Fig. 2b, c and Supplementary Data 2).

## Baseline PAM50 subtype and differentially expressed genes in relation to ANA/PAL response

Baseline (BL) RNA-seq was performed for 20 patients (13 Sensitive and 7 Resistant) with sufficient tumor material (Supplementary Fig. 3a) and subjected to PAM50 subtype determination. Two additional patients had BL tumor RNA-seq-derived PAM50 subtype previously reported[19]. There were 9 Luminal A (LumA), 8 LumB, and 5 non-luminal BCs (3 Basal-like, 2 HER2E) (Fig. 2A). PAM50 subtype was significantly associated with ANA/PAL response (p < 0.001, Table 1), with resistance

finding no significant differences. While PIK3CA mutations were common in Sensitive cases, this was not statistically significant (8/18 vs 1/8, P = 0.19). Assessment of CNV identified frequent amplifications in MYC[18] and CCNE2, consistent with ET-resistant phenotype

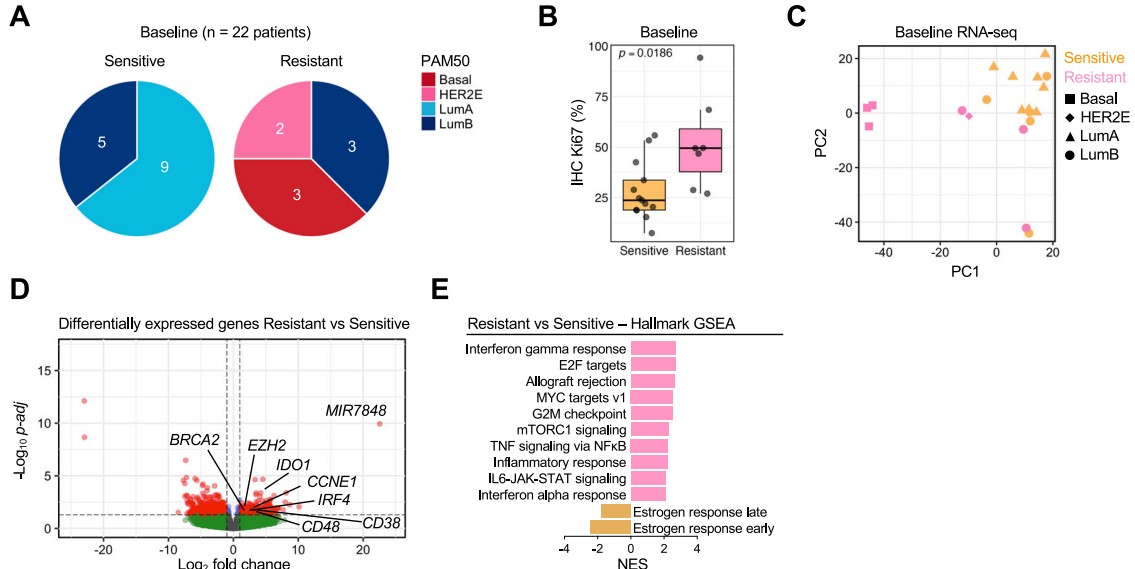

**Fig. 2 | Baseline comparison of PAM50 subtype and differentially expressed genes and pathways between ANA/PAL-sensitive and ANA/PAL-resistant cases.** **A** composition of PAM50 subtypes of ANA/PAL-Sensitive or -Resistant tumors. **B** IHC Ki67 of Sensitive ($n$ = 13 samples) and Resistant tumors ($n$ = 7 samples). Statistics calculated with two-sided Mann–Whitney $U$ test. Boxplot shows the median (center line), the 25th and 75th percentiles (lower and upper box bounds), and whiskers extending to the most extreme data points within 1.5× the interquartile range. **C** Principal component analysis (PCA) and **D** volcano plot of differentially expressed genes by DEseq2. −Log10 $p$-adj values are plotted on the $y$-axis. **E** Top oncogenic pathways significantly up or downregulated by Hallmark GSEA analysis in Resistant vs Sensitive tumors.

observed in all 5 non-luminal, 3 of 8 LumB, and none of the 9 LumA. This is consistent with the higher Ki67 observed in Resistant BCs (Fig. 2B and Table 1).

BL RNA-seq analyses revealed 630 DEGs (FDR adj. $p \leq 0.05$) between Sensitive and Resistant cases. Principal component analysis (PCA) of DEGs demonstrated clustering of Sensitive BCs, while Resistant BCs were scattered, with notable tight clustering of the 3 Basal-like BCs (Fig. 2C). Statistically significant, positively enriched DEGs of interest in Resistant BCs included tumor-associated effectors *CCNE1*, *IDO1*, *IRF4*, *BRCA2*, *EZH2*, and immune cell markers such as *CD38* and *CD48* (Fig. 2D), among others (Supplementary Data 3). Top upregulated signaling pathways in Resistant BCs included interferon (IFN) γ signaling, E2F targets, MYC targets, G2M checkpoint, mTORC1, TNF signaling, inflammatory response, IL6-JAK-STAT, and IFNα signaling (Fig. 2E), highlighting oncogenic pathways that govern proliferative advantage and inflammatory response. Estrogen response pathways were significantly downregulated in Resistant BCs (Fig. 2E). To support these findings, we analyzed the NeoPalAna initial cohort (NeoPalAna-Initial)[11], which showed consistent shared DEGs and dysregulated oncogenic pathways and estrogen signaling in Resistant tumors (Supplementary Fig. 4a, b). These data reveal intrinsic tumor dependencies impacting response to CDK4/6 inhibitors.

## C1D15 on-treatment PAM50 subtype and differentially expressed genes and proteins in relation to ANA/PAL response

Research tumor biopsies collected at C1D15 with adequate tumor content were subjected to RNA-seq (FFPE core, $n$ = 28: 19 Sensitive and 9 Resistant) and mass spectrometry (OCT frozen core, $n$ = 17: 12 Sensitive and 5 Resistant; Supplementary Fig. 3b–d) to assess transcriptomic and proteomic differences between Resistant and Sensitive tumors. PAM50 subtype determination based on C1D15 RNA-seq demonstrated the 19 Sensitive cases being predominantly LumA ($n$ = 18) or normal ($n$ = 1), while the 9 Resistant cases included 3 LumA, 2 LumB, 1 HER2E, and 3 Basal-like BCs (Fig. 3A). Like BL, PAM50 subtype at C1D15 was significantly associated with Resistance ($p$ = 0.0014). Figure 3B shows the Ki67$_{C1D15}$ distribution of Sensitive vs Resistant cases.

RNA-seq at C1D15 revealed 1,148 DEGs (FDR adj. $p \leq 0.05$) between Sensitive and Resistant cases (Supplementary Data 4). Similar to BL findings, PCA of DEGs at C1D15 showed a distinct distribution of Sensitive vs Resistant cases and clustering of Basal-like BCs (Fig. 3C). Top upregulated genes in Resistant tumors included *TOP2A, E2Fs, CCNE1, BUB1, AURKB*, and *STMN1* (Fig. 3D), reflecting their high proliferative state. Hallmark GSEA demonstrated enrichment of pathways such as E2F targets, G2M checkpoint, MYC targets, mTORC1 signaling, mitotic spindle, DNA repair, and IFN signaling, and lower expression of pathways related to estrogen response, TGFβ signaling, myogenesis, and epithelial-mesenchymal transition in Resistant tumors (Fig. 3E and Supplementary Fig. 5).

To identify genes associated with proliferation despite ANA/PAL treatment in Resistant BCs, we analyzed C1D15 RNA-seq data and compared genes correlating with Ki67$_{C1D15}$ levels between Resistant and Sensitive tumors (Fig. 3F and Supplementary Table 5). As expected, Ki67$_{C1D15}$ levels in Resistant tumors were positively correlated with *MYC*, cell cycle genes (*CCNE1, CCNA1, CCNA2, CDK6, PLK1*), and translation factors (*EIF4EBP1, EIF4E2, E2F2*), consistent with their highly proliferative state, and negatively correlated with *ESR1* and *AR*, suggesting a less dependence on hormone receptor-mediated signaling. Interestingly, Ki67$_{C1D15}$ levels also positively correlated with genes involved in DNA repair (*BRCA2*), IFN and immune responses (*ADAR, IDO1, CD274* [PD-L1]), and inflammation (*IRAK1, IL1A*) in Resistant tumors (Fig. 3F), suggesting potential relevance of these pathways in ANA/PAL-resistant tumors.

Proteomic analysis confirmed genes and pathways associated with resistance, similar to the transcriptional findings. Figure 3G highlighted significantly upregulated proteins important for cell cycle progression (CDK2, CDK4), mTOR signaling and translation (RPS6, RPS6KB1, EIF4A1), interferon and inflammatory mediators (ADAR, STAT1, STAT3, IRAK1) in Resistant tumors (and Fig. 3H). The dysregulated pathways in Resistant tumors showed high correlation between transcriptomes and proteomes (Spearman correlation $r$ = 0.92) (Fig. 3I). Among the Hallmark pathways of interest, high RNA-protein correlations were observed in interferon and

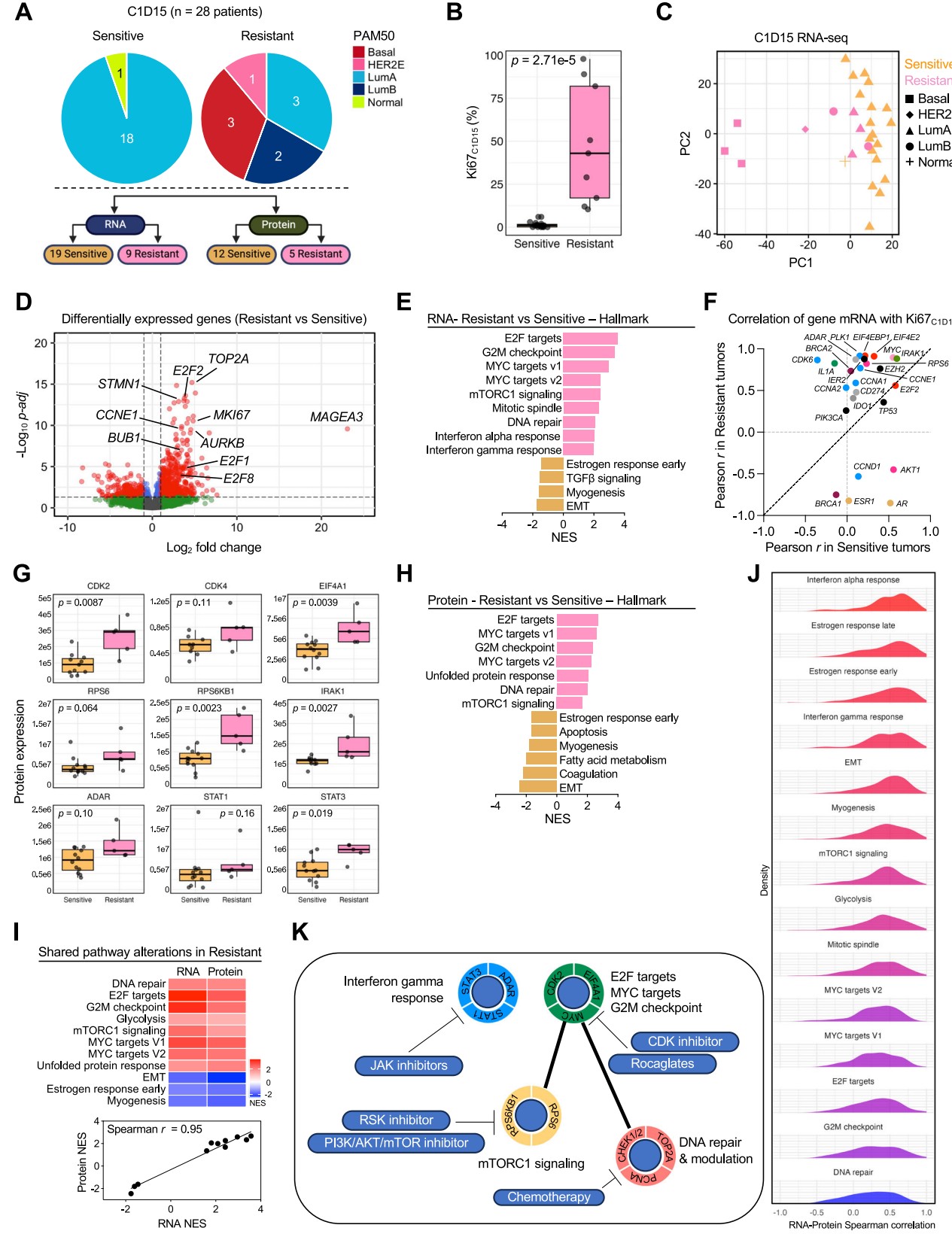

estrogen response pathways, while more discordance was seen in MYC, cell cycle, and DNA repair pathways (Fig. 3J), which may suggest post-transcriptional or post-translational processes in response to ANA/PAL. These integrated analyses highlighted potential oncogenic pathways as therapeutic targets for further investigation (Fig. 3K).

We then assessed treatment-induced changes at C1D15 relative to BL. In cases with paired samples, subtype switching to LumA was uniformly observed in Sensitive cases (BL LumB, $n = 5$), and in 2 Resistant BCs (1 LumB, and 1 HER2E at BL) (Fig. 4A). One Resistant LumB switched to HER2E. All 3 Basal-like BCs remained Basal-like at C1D15. Among significant DEGs ($\geq$twofold changes) between BL and

**Fig. 3 | Transcriptomic and proteomic analysis of ANA/PAL-Sensitive and ANA/PAL-Resistant cases at C1D15. A** Schematic of samples analyzed by RNA-sequencing and mass spectrometry. **B** Ki67$_{C1D15}$ of ANA/PAL-Sensitive ($n = 19$ samples) or -Resistant tumors ($n = 9$ samples). Statistics calculated with two-tailed Mann–Whitney $U$ test. **C** PCA and **D** volcano plot of differentially expressed genes by DEseq2. −Log10 $p$-adj values are plotted on the $y$-axis, and **E** top oncogenic pathways significantly up or downregulated by Hallmark GSEA analysis in Resistant cases are shown. NES Normalized enrichment score. **F** Pearson correlation of Ki67$_{C1D15}$ and mRNA expression of genes of interest in Resistant ($y$-axis) versus Sensitive ($x$-axis) tumors. **G** Box plots of relevant protein effectors involved in cell cycle, PI3K/AKT/mTOR, and interferon pathways in Sensitive ($n = 12$ samples) or

Resistant ($n = 5$ samples). Statistics calculated with two-tailed Mann–Whitney $U$ test. As also in (**B**), boxplots show the median (center line), the 25th and 75th percentiles (lower and upper box bounds), and whiskers extending to the most extreme data points within 1.5× the interquartile range. **H** Hallmark pathway analysis of protein expression in Sensitive and Resistant tumors. **I** Hallmark pathway alterations shared at RNA and protein levels in Resistant versus Sensitive tumors. **J** Spearman correlation between RNA and protein expression in 14 paired samples. Candidate genes/proteins belonging to assigned Hallmark pathways were plotted. **K** Summary schematic of enriched oncogenic pathways at RNA and/or protein level in Resistant tumors at C1D15 with identified targetable effectors of interest and relevant inhibitors.

C1D15, 3032 genes were uniquely upregulated in Sensitive cases and 735 in Resistant cases, with 1208 shared. For downregulated genes, 909 were unique to Sensitive cases and 613 to Resistant cases, with 591 shared (Fig. 4B and Supplementary Data 6). Key cell cycle genes (including *CCNB2, AURKB, E2F1* and *E2F2)* were suppressed at C1D15 in Sensitive, but not Resistant samples (Fig. 4C). While ANA/PAL treatment caused similar overall changes in both Sensitive and Resistant tumors (Fig. 4D), key pathways like cell cycle and interferon signaling, though suppressed at C1D15 compared to BL, remained relatively elevated in Resistant tumors (Fig. 4E). Persistent gene expression and pathway activation despite ANA/PAL establish these modules as prominent resistance mechanisms.

### Targeting hyperactive interferon and inflammatory signaling with JAK inhibitors in CDK4/6i-resistant models

To examine the functional relevance of candidate pathways observed in the ANA/PAL-resistant tumors, we generated MCF7 Palbo-R and Abema-R through long-term dose-escalation culture of MCF7 cells in palbociclib and abemaciclib, respectively (Fig. 5A, B). Compared to parental cells, MCF7 Palbo-R and Abema-R showed upregulated cell cycle effectors (i.e., CDKs and cyclins), and enrichment of interferon and TNF signaling by RNA-seq analysis (Fig. 5C–F), consistent with that observed in the ANA/PAL-resistant tumors (Figs. 2E and 3E). Upregulated IFN/inflammatory signaling was confirmed by increased STAT3 phosphorylation and elevated levels of the downstream interferon-stimulated gene 15 on immunoblot (Fig. 5G). Notably, RB expression was retained, as its loss is a known mechanism of CDK4/6i resistance in cell line models[20]. Elevated AKT phosphorylation was also observed, supporting augmented PI3K/AKT signaling in resistant lines consistent with previous literature[21].

We then used these resistant models to assess the effects of compounds targeting the JAK-STAT pathway, the downstream mediators of IFN signaling, and inflammatory pathways. We chose to investigate JAK1/2 inhibitors as they are already clinically approved to treat inflammatory, autoimmune, and neoplastic diseases such as myeloproliferative neoplasms, including ruxolitinib (JAK1/2), fedratinib (JAK2/FLT3), momelotinib (JAK1/2/ACVR1), and pacritinib (JAK2/ACVR1/IRAK1)[22,23], and also itacitinib (JAK1), currently under phase 3 investigation for graft-versus-host disease[24]. Screening with these inhibitors in parental MCF7 and CDK4/6i-resistant cells revealed similar sensitivity across the cell lines, with pacritinib showing the highest anti-cancer potency (Fig. 5H), suggesting that dual inhibition of JAK2 and IRAK1, which targets interferon and TNF signaling via NFKB pathways, could be beneficial. In Palbo-R cells, pacritinib alone effectively suppressed STAT3 phosphorylation and downregulated cyclin D1, with additive anti-cancer effects when combined with palbociclib (Fig. 5I, J). In contrast, combining pacritinib with abemaciclib was necessary to effectively reduce pSTAT3 and Cyclin D1, possibility explaining the synergistic anti-cancer effect in Abema-R cells (Fig. 5I, J). We lastly tested pacritinib and palbociclib in ER + /HER2− patient-derived xenograft (PDX) organoid models. WHIM58 PDX was derived

from a metastatic sternal lesion and harbors hotspot mutations in *AKT1* and *TP53*, in addition to having *MYC* and *CCND1* amplifications. WHIM43 was derived from a metastatic rib lesion and was found to have *RB1*-loss by WES and negative IHC staining of RB[25]. Both organoid models showed reduced sensitivity to palbociclib (Fig. 5K) but were responsive to pacritinib at low micromolar doses (Fig. 5L), which was comparable to the response observed in MCF7 Palbo-R/Abema-R cells. These data support the potential importance of JAK/STAT signaling in CDK4/6i resistant BCs.

### BL and treatment-induced changes in immune cell populations

The upregulation of IFN signaling in both CDK4/6i-resistant cell lines and ANA/PAL-Resistant BCs, along with elevated markers of immune cells and immune checkpoints in Resistant tumors at baseline (Fig. 2) and at C1D15 (Fig. 3), prompted further exploration of the tumor immune microenvironment. Using CIBERSORT analysis, we deconvoluted bulk RNA-seq data to estimate immune cell type fractions at BL and C1D15 (Fig. 6A). Compared to Sensitive tumors, Resistant tumors at BL exhibited a significantly higher abundance of immune cells, including CD8 + T cells and M1 macrophages (Fig. 6B). ANA/PAL treatment increased several immune cell types (CD8 T cells, activated NK cells, plasma cells, macrophages, and monocytes) in Sensitive but not Resistant tumors (Fig. 6B), suggesting a more favorable anti-tumor immune microenvironment in Sensitive tumors.

We further analyzed BL RNA-seq data using an 18-gene T-cell-inflamed TME signature[26], previously linked to poor PFS benefit from PAL in metastatic ER+/HER− BC in the PALOMA-2 trial[27,28]. We found significant upregulation of this signature in resistant tumors in Neo-PalAna ET-R (Fig. 6C, D). This, along with a higher abundance of immune cells, including CD8 + T cells and M1 macrophages from CIBERSORT analysis at BL, suggests a pre-existing inflamed but immune suppressive microenvironment in Resistant tumors.

### Enhanced IDO1 expression in resistant tumors and association with PAM50 subtype

We further assessed immune checkpoints (ICs) expression in Resistant tumors. There was a significant correlation between ICs, including *IDO1* and *CD274*, and Ki67, suggesting a proliferative role (Fig. 7A). IDO1 canonically suppresses anti-tumor immunity through tryptophan depletion[28,29] and was part of the 18-gene immune signature (Fig. 6C). *IDO1* was also one of the top upregulated genes in Resistant tumors in BL RNA-seq (Figs. 7B and 2D and Supplementary Fig. 4a). We further performed immunohistochemistry (IHC) for IDO1 on BL biopsies from the NeoPalAna ET-R cohort, which demonstrated elevated expression in Resistant versus Sensitive tumors (Fig. 7C). IDO1 IHC, quantified with Allred score, modestly correlated with mRNA levels (Fig. 7D). Using a 1% cutoff, all 4 Basal-like BCs were IDO1 positive, compared to 5 of 8 LumB and 2 of 8 LumA BCs (Fig. 7E). Basal-like BCs had the highest IDO1 mRNA expression compared to LumB and LumA subtypes (Fig. 7F). The correlation was confirmed in the TCGA BRCA and METABRIC cohorts (Fig. 7G).

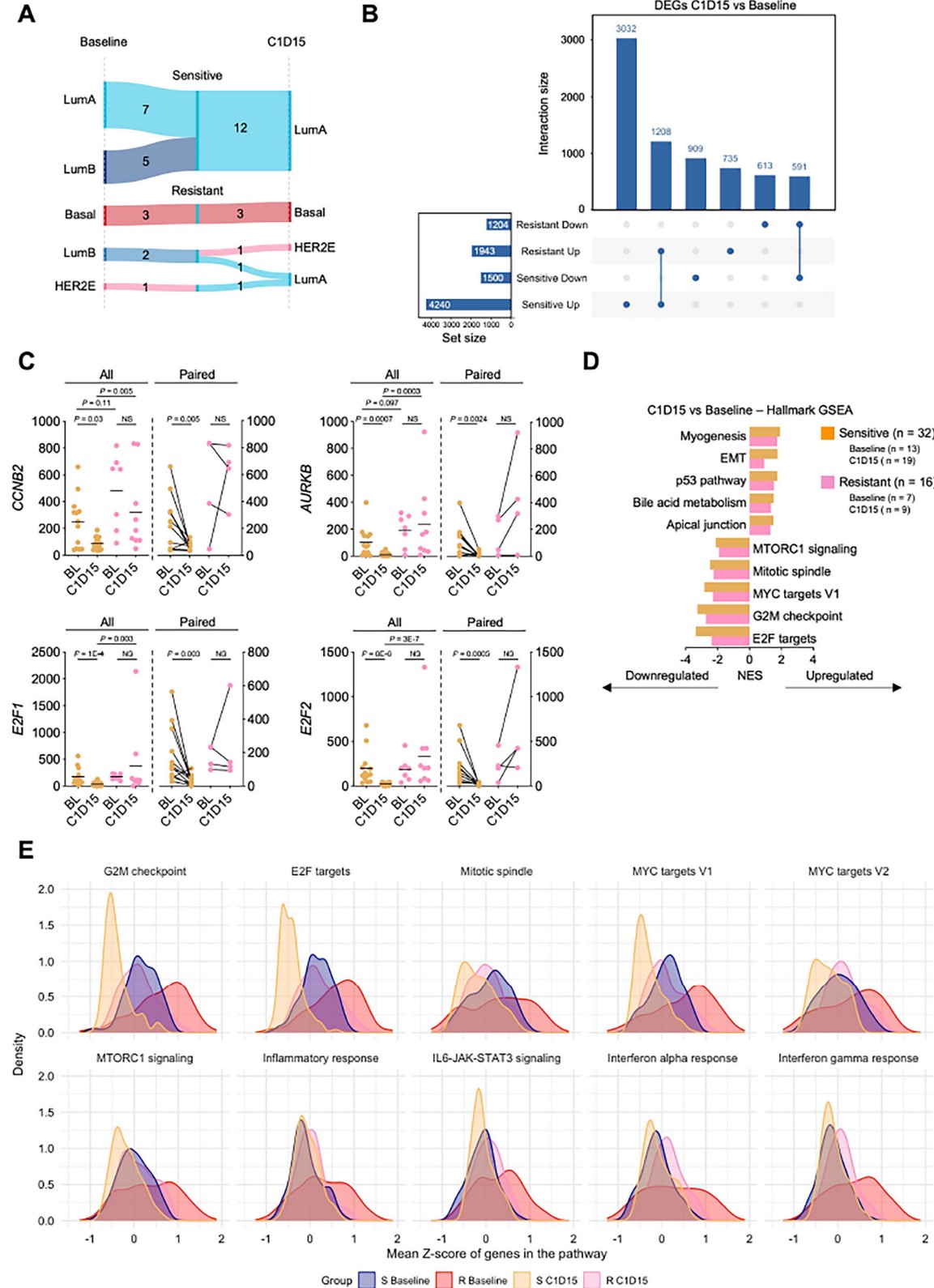

**Fig. 4 | Treatment induced transcriptomic changes. A** Alterations in PAM50 subtypes throughout treatment. **B** Unique and shared, upregulated and downregulated, DEGs (adj. *P*-value < 0.05 and log twofold change > 1 or < −1) across C1D15 versus baseline timepoints in Resistant and Sensitive patients. **C** Notable cell cycle genes significantly downregulated at C1D15 versus baseline in Sensitive but not Resistant samples. Left panels represent all samples at each timepoint with two-tailed Mann–Whitney *U* test; *n* = 13 baseline S, 19 C1D15 S 7 baseline R, 9 C1D15 R. Dashed line denoting mean expression. Right panels represent paired samples throughout treatment with Wilcoxon match-pairs signed rank test; *n* = 12 S and 4 R. **D** Hallmark GSEA analysis of C1D15 versus Baseline samples in Sensitive and Resistant tumors. **E** Density plot of genes in pathways of interest. Z-scores were calculated for each gene across Sensitive and Resistant tumors at Baseline and C1D15 timepoints followed by plotting of mean group expression of each gene.

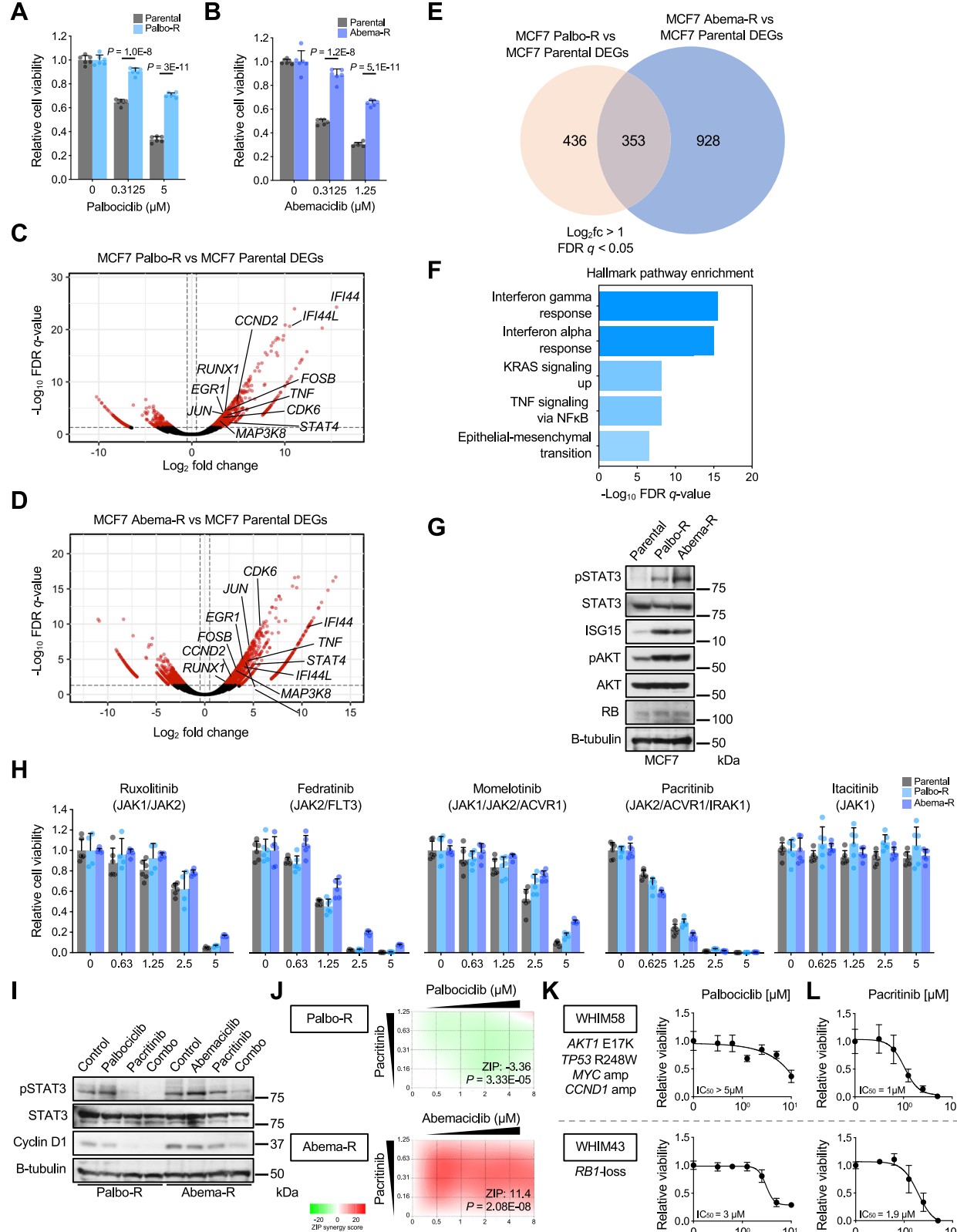

## NeoPalAna 33-gene resistance signature is prognostic in metastatic ER+HER− breast cancer receiving ET/CDK4/6i

To validate BL ANA/PAL response biomarkers, we derived a resistance gene signature using BL RNA-seq data from NeoPalAna ET-R and previous microarray data from NeoPalAna Initial to increase sample size (Fig. 8A and Supplementary Fig. 6). Logistic regression models were performed for receiver operating characteristic (ROC) curve analysis

for candidate genes across both training cohorts. The 33-gene signature includes key regulators of cell cycle, oncogenic signaling, proliferation, transcription, DNA repair, metabolism, and immune and interferon response mediators (Fig. 8B). PCA of the gene signature effectively separated Resistant and Sensitive tumors (Fig. 8C, D). The gene signature outperformed Ki67, with an ROC curve area under the curve (AUC) of 0.967 (95% CI 0.902–1) in NeoPalAna ET-R and 0.992

**Fig. 5 | Targeting hyperactive interferon and inflammatory signaling with JAK inhibitors in CDK4/6i-resistant models. A** Cell viability assays of parental and generated Palbo-R and **B** Abema-R MCF7 cells. Cells were treated at the indicated palbociclib or abemaciclib doses for 7 days. $n = 6$ biological replicates per group at each treatment condition. Data are presented as mean values +/− SD. Statistics assessed by two-tailed Student's $t$-test. **C** Volcano plots of DEGs of Palbo-R and **D** Abema-R compared to parental MCF7 cells. **E** Unique and shared DEGs in Palbo-R and Abema-R versus parental MCF7 cells with a log$_2$ fold change greater than 1 and false discovery rate $q$-value less than 0.05. **F** Hallmark pathway enrichment analysis of DEGs in (E). **G** Immunoblot of parental, Palbo-R, and Abema-R MCF7 cells. Immunoblots are representative of two independent experiments. **H** Cell viability assays of parental, Palbo-R, and Abema-R MCF7 cells treated at the indicated JAK1/2 inhibitor micromolar doses for 6 days. $n = 6$ biological replicates per group at each

treatment condition except for MCF7 Palbo-R cells treated with ruxolibinitb ($n = 4$ biological replicates per group at each treatment condition). Data are presented as mean values +/− SD. **I** Immunoblot of Palbo-R and Abema-R MCF7 cells treated with CDK4/6 inhibitor (2 μM palbociclib, 1 μM abemaciclib), 1 μM pacritinib, or combination. Cells were treated for 36 hours. Immunoblots are representative of two independent experiments. **J** Zero Interaction Potency (ZIP) drug synergy calculations of CDK4/6i and pacritinib at indicated doses, utilizing a twofold dilution ratio. Green represents a negative ZIP score (antagonism when ZIP < −10) and red represents a positive ZIP score (synergism when ZIP > 10). Calculations performed with SynergyFinder. **K** Cell viability assays of two ER+/HER2− PDX organoid models. Organoids were treated for 5 days at the indicated drug doses of palbociclib ($n = 8$ biological replicates) and **L** pacritinib ($n = 4$ biological replicates). Data in (**K**) and (**L**) are presented as mean values +/− SD.

(95% CI 0.971−1) in NeoPalAna Initial (Fig. 8E, F and Supplementary Data 7), with minimal improvement when Ki67 was added. We performed bootstrap analysis[30] in NeoPalAna ET-R to estimate variability and correct for potential bias in our results, which remained high for Ki67 (corrected AUC = 0.818), the signature alone (corrected AUC = 0.965, individual gene corrections in Supplementary Data 8), or in combination with Ki67 (corrected AUC = 0.963).

For independent validation, we examined a "Real-world" external cohort with RNA-seq analysis using the HTG EdgeSeq Oncology Biomarker Panel ($n = 2560$ genes) on pre-treatment samples with clinical outcomes for 151 metastatic ER+/HER2− BC patients treated with first-line CDK4/6i (~90% palbociclib) and either an aromatase inhibitor (AI) ($n = 115$) or fulvestrant ($n = 36$)[31]. Blinded to clinical outcomes, we applied the NeoPalAna resistance gene signature to the RNA-seq data, stratifying patients into high, medium, or low score groups by expression score tertiles. Kaplan−Meier (K−M) analyses subsequently performed showed higher gene signature scores significantly associated with shorter progression-free survival (PFS) (Hazard Ratio = 2, $p = 0.012$, median PFS: Low = 30.18 months, Medium = 22.03 months, High = 18.21 months) in the overall population (Fig. 8G). This association was also significant with AI and a trend with fulvestrant. Higher scores were also associated with a shorter overall survival (OS) across this cohort (HR = 2, $p = 0.03$) (Fig. 8H). These data provided further validation of biomarkers of resistance from the NeoPalAna cohorts.

## Discussion

CDK4/6is have revolutionized the treatment landscape for HR+/HER2− BCs[32,33]. However, not all patients benefit from CDK4/6i. Standard-of-care options, including selective estrogen receptor degrader[34], PI3Ki[35], AKTi[36], and mTORi[37], have shown limited efficacy in CDK4/6i resistant BCs[38]. Improved understanding of resistance mechanisms and biomarkers of response to ET/CDK4/6i is necessary.

Results from the NeoPalAna ET-R cohort confirmed our prior finding[11] that the addition of PAL to ANA effectively suppresses cell proliferation in a subset of BCs resistant to ANA monotherapy, achieving CCCA in 57.6% (95% CI: 39.2−74.5%). Analysis of the NeoPalAna ET-R cohort showed that Resistant BCs are more likely to be grade 3, Ki67-elevated, non-luminal subtype, and accompanied by upregulated cell cycle genes, mTOR, IFN signaling, inflammatory response, and immune checkpoint expression at BL.

Comparing Sensitive to Resistant tumors, we found no significant association with mutations in *TP53* and *PIK3CA*, or CNVs of cell cycle mediators analyzed, aligning with the literature on CDK4/6i efficacy across various mutation profiles. A lower incidence of *PIK3CA* mutation in Resistant cases (1/8) versus Sensitive cases (8/18) was noted, consistent with the NeoPalAna Initial cohort[11]. This is potentially explained by the association of *PIK3CA* mutation with LumA subtype in the early stage setting[13]. We observed CNVs in DNA repair and PI3K/AKT/mTOR pathways being enriched in Resistant cases. However, these results should be interpreted with caution due to the small sample size.

Upregulation of cell cycle genes and pathways, including *CCNE1*, *MYC*, and G2/M, are consistent with findings from NeoPalAna Initial and shorter PFS in patients with metastatic BC receiving ET/PAL[27,31,39−41]. *CCNE1* mRNA expression is higher in LumB and non-luminal subtypes compared with LumA[39,40] and Cyclin E1 overexpression is common at developing resistance to PAL in ER + BC cell lines[42], possibly through activating CDK2[41]. MYC overexpression can sustain DNA synthesis during CDK4/6i[41] and promote pRB1 degradation[43]. Interestingly, MYC requires CCNE1 to escape from PAL-induced growth arrest[44]. Therapeutic opportunities targeting cell cycle mediators CDK2[41,45], CDK7[46], eIF4A[20,47], PLK1[40], PKMYT1[48], AURKA[49], and MYC are being explored.

We observed higher mTORC1 signaling in Resistant tumors, consistent with studies in the metastatic setting, including PALOMA-2 and PALOMA-3[27]. The crosstalk between the PI3K/AKT/mTOR pathway and Cyclin D1/CDK4/6/Rb is well established[50−52] and combined inhibition of PI3K/mTOR and CDK4/6 enhances anti-tumor effects in preclinical models[21,53,54] and in INAVO120 trial[55]. The potential of inhibitors against S6/RSK signaling (PMD-026)[56] in treating ER + MBC is also being evaluated in an ongoing phase I/II trial (NCT 04115306).

ANA/PAL-resistance observed in non-luminal BCs aligns with the negative association with estrogen response genes, and may links observations between higher ER/estrogen response signaling and longer PFS in metastatic patients receiving CDK4/6i[27,31,57]. In the metastatic setting, while CDK4/6i benefit were observed across subtypes[39,58,59], HER2E and Basal-like subtypes consistently show shorter PFS for metastatic patients on CDK4/6i/ET[31,39,40,58,59].

The enrichment of interferon and inflammatory signaling in CDK4/6i resistant BCs in both pre-treatment and C1D15 biopsies is consistent with previous studies[41,60,61]. STAT3 inhibitors have been shown to overcome CDK4/6i resistance[61], though they are still in early clinical development[62]. Our studies on resistant cell line and PDX organoids demonstrated, for the first time, that clinical-grade JAK2 inhibitors, particularly pacritinib (which also inhibits IRAK1), could be therapeutic for CDK4/6i resistant BCs. Anti-breast cancer efficacy was observed at approximately 1 μM pacritinib, a concentration below the clinical steady-state levels achieved with the standard human dose administered for myeloproliferative disease treatment[63]. This suggests that a similarly tolerable dosing regimen could be applicable for breast cancer patients, though further in vivo studies are warranted. Notably, we did not observe differences in the sensitivity of JAK2 inhibitors across the parental and Palbo-R/Abema-R MCF7 cell lines. These data suggest JAK-STAT may not be the sole driver mediating drug resistance and is likely also a consequence of other signaling pathways. This may be particularly the case in the primary breast tumor setting, where interactions with the tumor microenvironment can significantly influence signaling dynamics and therapeutic response. Further studies are warranted to better elucidate the role of JAK/STAT pathway upregulation in CDK4/6i resistant breast cancer.

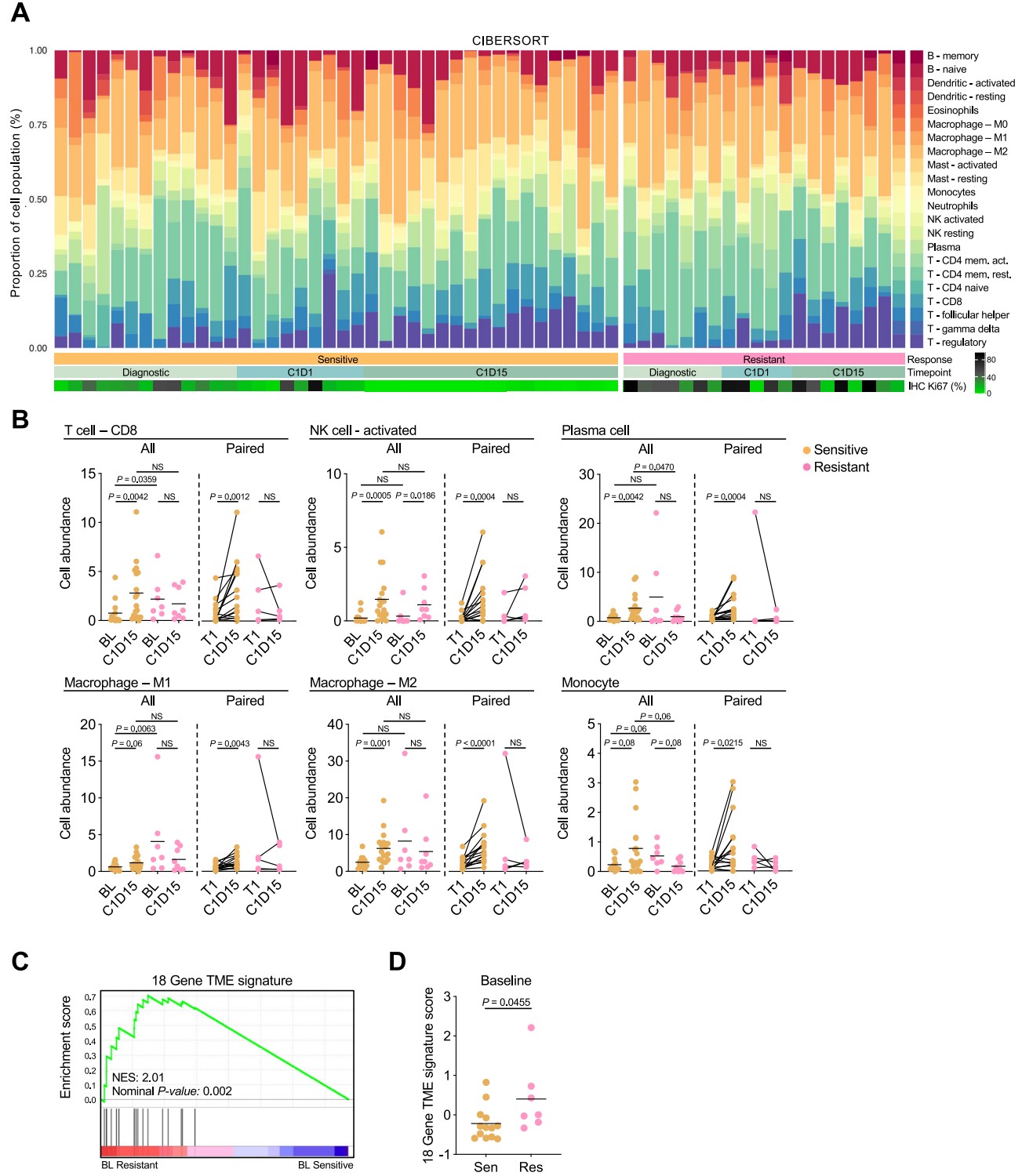

**Fig. 6 | Baseline and treatment-induced changes in immune cell population.**
**A** CIBERSORT analysis of cell populations in sensitive and resistant tumors across treatment timepoints. **B** Abundance of cell populations, including those mediating anti-tumor responses. Diagnostic samples included either baseline or C1D1 samples. Left panels represent all samples at each timepoint with two-tailed Mann–Whitney *U* test; *n* = 13 baseline S, 18 C1D15 S, 7 baseline R, 8 C1D15 R. Right panels represent paired samples throughout treatment with Wilcoxon match-pairs signed rank test; *n* = 15 S and 5 R. **C** Enrichment plot of an 18 gene T-cell-inflamed

tumor microenvironment (TME) signature described by Ayers et al. The 18 genes include: *CD274, CD276, CCL5, CD27, CD8A, CMKLR1, CXCL9, CXCR6, IDO1, LAG3, NKG7, PDCD1LG2, PSMB10, HLA-DQA1, HLA-DRB1, HLA-E, STAT1*, and *TIGIT*. NES: Normalized enrichment score. Nominal *p*-value calculated by GSEA. **D** 18 Gene TME signature score calculated per each baseline tumor sample. Row z-scores were calculated for each gene, which were summed together. Statistics by two-tailed Student's *t* test; *n* = 13 Sensitive and 7 Resistant samples.

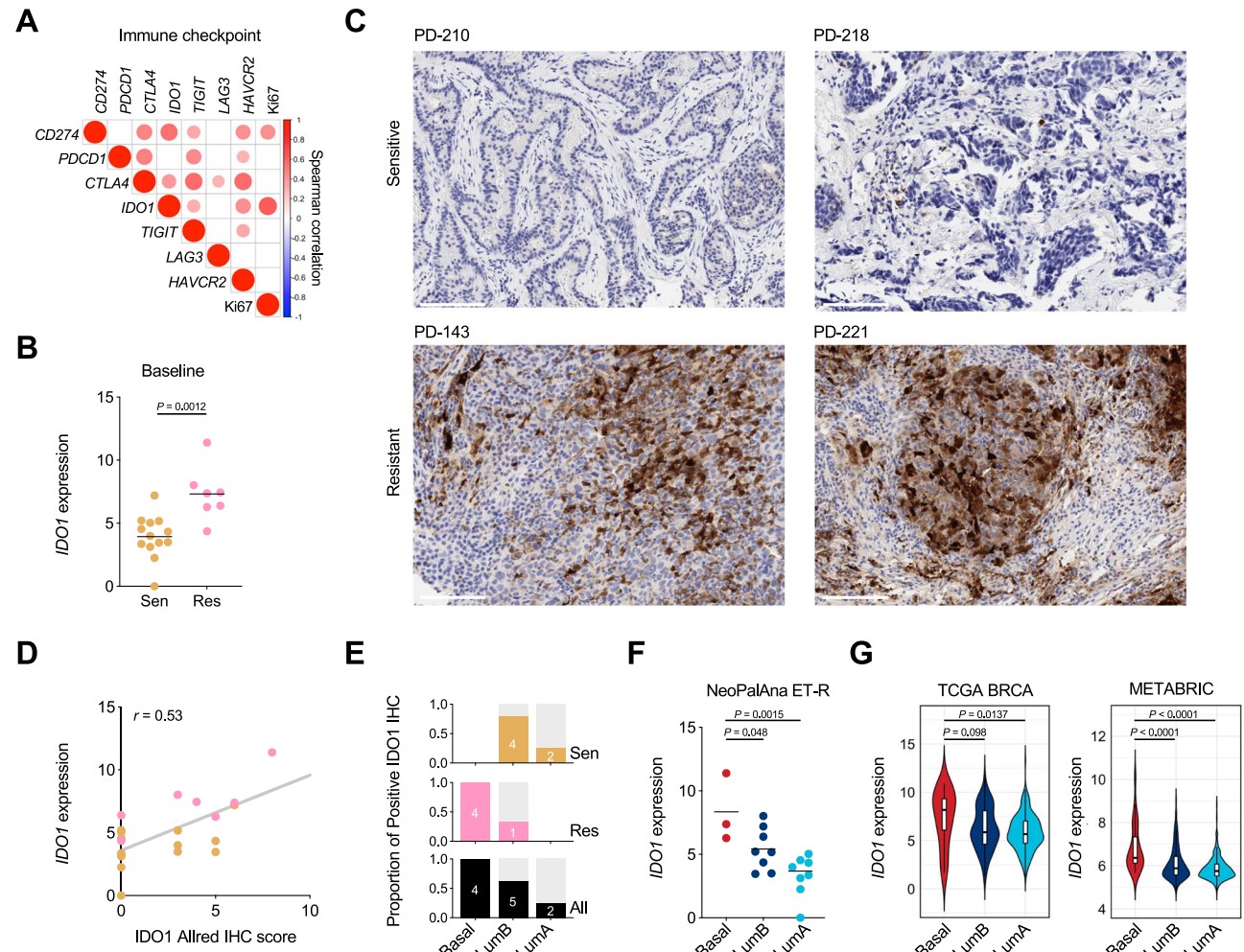

**Fig. 7 | Enhanced IDO1 expression in Resistant tumors and associated with PAM50 subtype. A** Spearman correlation of Ki67 and mRNA expression of immune checkpoint genes from NeoPalAna ER tumors ($n = 63$ samples). The size of the circles positively scale with the Spearman correlation. **B** IDO1 expression in Sensitive and Resistant tumors at baseline. Expression denoted by $Log_2 (x + 1)$. $n = 13$ Sensitive and 7 Resistant samples. Statistics by two-tailed Student's $t$ test. **C** IDO1 IHC staining in two Sensitive and two Resistant tumors. Scale bar: 100 μM. **D** Pearson correlation between IDO1 quantification at mRNA by RNA-seq and at protein by IHC in Sensitive ($n = 13$) and Resistant ($n = 7$) tumors. **E** Proportion of IDO1 IHC positivity by PAM50 subtype and ANA/PAL response. **F** IDO1 expression

across PAM50 subtypes from baseline tumors from NeoPalAna ET-R. $n = 3$ Basal, 8 Luminal B, and 8 Luminal A. 12 Sensitive and 7 Resistant samples. Statistics by one-way ANOVA comparing to Basal. **G** Violin plots of IDO1 expression across PAM50 subtypes from TCGA BRCA (left, tumor samples: Basal $n = 12$, LumB $n = 93$, LumA $n = 271$) and METABRIC (right, tumor samples: Basal $n = 29$. LumB $n = 411$, LumA $n = 656$). Statistics by one-way ANOVA comparing to Basal. Plots show the median (center line), the 25th and 75th percentiles (lower and upper box bounds), and whiskers extending to the most extreme data points within 1.5× the interquartile range.

ANA/PAL-resistant BCs showed increased ICs, including IDO1 expression, and the 18-gene immune gene signature. CIBERSORT analysis indicated a higher abundance of immune cells, including CD8 + T cells and macrophages in Resistant versus Sensitive tumors at BL. This suggests an interplay between ET/CDK4/6i resistance and a dysregulated tumor immune microenvironment in ER + /HER2- BC. IFNγ is a canonical activator of JAK-STAT signaling and it has been shown that IFNγ drives PD-L1 expression[64] and upregulates other suppressive molecules like IDO1[65]. An IFN palbociclib resistance signature (IRPS) was linked to IC expression and immunosuppressive tumor microenvironments in TCGA and METABRIC ER+ BCs[60], implying a connection between tumor intrinsic IFN pathway activation and an inflamed but suppressive microenvironment. Additionally, resistance to ET was linked to overexpression of ICs (IDO1, LAG3, PD1), the LumB subtype, and STAT1 expression in NET trials[66]. Similar observations were made in PALOMA-2, where PD1 expression was a relative PAL resistant biomarker[57]. PALOMA-2[27] and PALOMA-3[41] trials also showed an upregulated IFN pathway and immune gene signature were

associated with shorter PFS[27]. This 18-gene T-cell–inflamed TME signature predicts IC inhibitor response[26,28], offering a potential therapeutic opportunity for these poor prognosis ET/CDK4/6i-resistant ER+/HER2− BCs. As such, efficacy by JAK-STAT suppression may be multidimensional through diminishing both oncogenic signaling and immune modulators such as PD-L1 and IDO1.

Our study has several limitations. It is a neoadjuvant study using on-treatment Ki67, a proliferation biomarker, to differentiate Sensitive versus Resistant BCs. Additionally, the relatively small sample size in our cohort, particularly in the number of Resistant patients, may reduce statistical power and increase the likelihood of false-positive and false-negative findings, and we acknowledge that interrogation of multiple variables could result in overfitting. We attempted to mitigate the challenges by including the NeoPalAna initial cohort as a second discovery cohort and through bootstrap analysis to identify potential biases and overoptimism. Importantly, the derived NeoPalAna resistant gene signature was prognostic for PFS in an independent cohort of metastatic ER+/HER2− BC patients on CDK4/6i/ET[31]. This provided

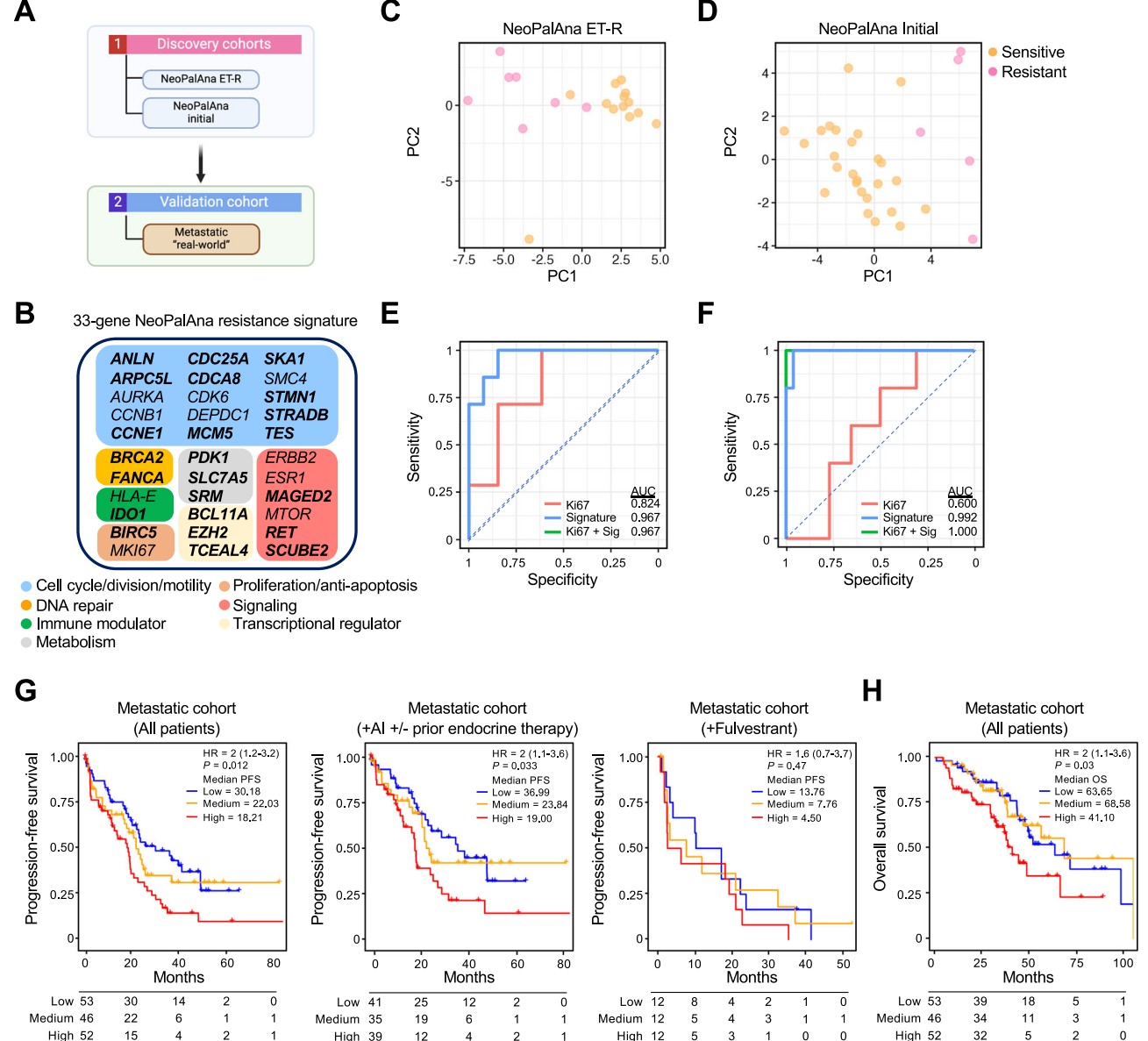

**Fig. 8 | NeoPalAna 33-gene signature predicts response to CDK4/6 inhibitors in primary and metastatic ER + HER- breast cancer. A** Schematic of discovery and validation cohorts. **B** 33-gene CDK4/6i resistance signature and annotated canonical, biological function. Bolded genes represent differentially expressed genes from the NeoPalAna ET-R cohort identified through DEseq2 analysis. **C** PCA of Sensitive and Resistant tumors by baseline gene expression signature in NeoPalAna ET-R (*n* = 13 Sensitive, 7 Resistant) and **D** NeoPalAna Initial (*n* = 26 Sensitive, 5 Resistant) cohorts. **E** Receiver Operating Characteristic (ROC) curves in NeoPalAna

ET-R and **F** NeoPalAna Initial of baseline Ki67, derived gene signature, and combination. **G** Kaplan–Meier progression-free survival (PFS) curves by gene signature score in all treated patients (left panel), those treated with CDK4/6i with aromatase inhibitors (middle panel), and those treated with CDK4/6i with fulvestrant (right panel) in the "Real world" Metastatic cohort. **H** Overall survival of all treated patients in the Metastatic cohort. Also in (**G**), HR Hazard Ratio, calculated with Log-Rank test, comparing to the low score cohort, with 95% confidence intervals.

validation of the resistance biomarkers identified in the NeoPalAna ET-R cohort and demonstrated, for the first time, that cell proliferation response in the neoadjuvant setting can inform biomarker development for CDK4/6 inhibitors. Considering the small sample size of the discovery cohort and the medium sample size of the independent validation cohort, future large studies are needed to further validate the signature and confirm these findings. Nonetheless, our findings may serve as a foundation for future investigations aimed at optimizing trial design in this cancer context and stratifying patients for targeted therapeutic strategies.

In summary, the NeoPalAna ET-R cohort showed that neoadjuvant ANA/PAL resulted in 58% CCCA in ET-resistant BC. Oncogenic pathways like cell cycle, mTOR signaling, IFN, and immune tolerance

markers, such as IDO1, were enriched in ANA/PAL-resistant tumors. ANA/PAL-resistance was associated with increased IC expression and an upregulated 18-gene T-cell-inflamed TME signature. Importantly, a 33-gene signature predicting the antiproliferative response to neoadjuvant ANA/PAL was prognostic for PFS on CDK4/6i/ET in an independent cohort of metastatic patients. These findings provide important insight into CDK4/6i resistance mechanisms and suggest novel therapeutic opportunities for treating CDK4/6i resistant BC.

## Methods
### Eligibility
Eligible patients included pre- and post-menopausal women ≥18 years, with a clinical stage II-III, ER+ (Allred score ≥3 or >1% ER positivity) and

HER2 (IHC 0 or 1+ or FISH negative) BC. Tumor Ki67 > 10% after ≥4 weeks of standard-of-care NET by central testing[67] was required. Additional key eligibility criteria included: Eastern Cooperative Oncology Group Performance Status 0–2, adequate organ and marrow function, and consenting to research access to archival tumor tissue from biopsies for diagnosis (pre-treatment) and Ki67 on standard-of-care NET prior to enrollment (C1D1). The study was approved by the Institutional Review Board at participating sites and followed the Declaration of Helsinki and Good Clinical Practice guidelines. Written informed consent was obtained.

## Study design and treatment

The primary endpoint was CCCA (Ki67$_{C1D15}$ ≤ 2.7%) on ANA/PAL at C1D15. A Simon optimal two stage phase II design was employed to allow a 90% chance of detecting a CCCA rate of ≥20% (alpha 0.1). The regimen was considered to have sufficient anti-tumor activity if 4 of 37 patients (stage 1, $n = 12$; stage 2, $n = 25$) achieved CCCA. The study was stopped after enrolling 34 patients as the primary endpoint was met, per the data safety and monitoring board.

Eligible patients started ANA (1 mg PO daily) and PAL (125 mg PO daily, days 1–21 of each 28-day cycle) on C1D1. Goserelin (3.6 mg SC every 28 days) was required if premenopausal. Tumor biopsy was performed on C1D15 for central Ki67 analysis. If Ki67$_{C1D15}$ > 10%, protocol therapy was discontinued due to inadequate response. Patients with Ki67$_{C1D15}$ ≤ 10% (or indeterminate) continued ANA/PAL for 5 cycles unless experiencing intolerable side effects, disease progression, or withdrawal from the study. Surgery occurred on Cycle 5 Days 11–13 with the last dose of ANA/PAL on the day prior to surgery. Clinical tape/caliper bi-dimensional tumor measurements and adverse events (AE) assessments by CTCAE 4.0 were performed on day 1 of each cycle and at the completion of 5 cycles of ANA/PAL. Clinical tumor response was assessed according to RECIST 1.1. The response rate was defined as the proportion of patients achieving complete or partial response after completion of 5 cycles of ANA/PAL, evaluated both among patients who completed therapy and, in the intent-to-treatment population.

## Ki67 immunohistochemistry (IHC) and Quantification

Ki67 testing on archival tissues from pre-treatment (baseline; BL) and C1D1 biopsies, and research C1D15 biopsies, was performed centrally in the Washington University AMP lab using the CONFIRM anti-Ki67 antibody (clone 30-9) and scored using pathologist-guided imaging analysis or point-counting as previously described[67].

## Formalin-fixed paraffin-embedded (FFPE) tumor RNA/DNA extraction

FFPE tumor blocks were sectioned and ≥50% tumor cellularity was required for RNA/DNA isolation. Microdissection was performed if overall tumor cellularity was <50%. Deparaffinization was then performed as previously described[68]. Nucleic acids were isolated using the Allprep DNA/RNA FFPE Kit (Qiagen, 80234). Germline DNA was extracted using the QIAamp DNA kit (Qiagen, 51304).

## RNA sequencing (RNA-seq) and analysis

Total RNA was prepared according to manufacturer's protocol using SeqPlex RNA Amplification Kit (Sigma), indexed, pooled, and sequenced on an Illumina NovaSeq 6000. Sequencing pipeline and data analysis were performed as previously described[69]. Differential gene expression analysis was conducted using DESeq2[70]. Gene set enrichment analysis (GSEA) was performed using GSEA software[71] (version 4.2.2) or pathway enrichment analysis at https://www.gsea-msigdb.org utilizing the Hallmark gene set (Supplementary Data 1). PAM50-based intrinsic subtype assignment was determined as previously described[72].

## Whole exome sequencing (WES)

Tumor DNA and matched leukocyte germline DNA, when available, were subjected to WES as previously described[73]. Somatic variants were called from WES tumor and normal paired BAMs using somatic-wrapper v2.2 (https://github.com/ding-lab/somaticwrapper), a pipeline designed for detection of somatic variants from tumor and normal exome data[73]. The pipeline merges and filters variant calls from four callers: Strelka (v2.9.2)[74], Mutect (v1.1.7)[75], VarScan (v2.3.8)[76], and Pindel (v0.2.5)[77]. Indel calls were obtained from Stralka2, Varscan, and Pindel. The following filters were applied to get variant calls of high confidence: normal VAF ≤ 0.02 and tumor VAF ≥ 0.05; rescue low-VAF variants in SMG gene list by removing tumor VAF ≥ 0.05 requirement; read depth in tumor ≥14 and normal ≥8; indel length <100 bp; all variants must be called by 2 or more callers; all variants must be exonic; and excluding variants in dbSNP but not in COSMIC. In cases where normal tissue counterparts were unavailable, somatic variants were called using the Mutect2 (v4.1.2.0) best-practice pipeline (https://gatk.broadinstitute.org). Silent mutations were not considered in the analysis. Somatic copy number variations (CNVs) were identified using our in-house GATK4 somatic CNV calling pipeline (https://github.com/gatk-workflows/gatk4-somatic-cnvs).

## Mass spectrometry proteomic analysis

Proteomic analysis was performed on 12 Sensitive and 5 Resistant patient samples. Frozen biopsy OCT block sections with tumor cellularity of ≥25% were processed and assayed on mass spectrometry as previously described[78]. Briefly, OCT samples were sectioned with interleaved collection of curls, resulting in equal distribution of material for RNA, Protein and DNA extracts. Curls were then washed with 70% ethanol, followed by washing with water and finally with 100% ethanol. Samples were lysed with 8 M Urea and sonicated with a Covaris S220 before being centrifuged. The resulting supernatant was used for downstream proteomics processing. Samples in urea were reduced and alkylated followed by trypsin digestion. The resulting peptides were then loaded onto Evotips and washed with 0.1% formic acid before being loaded onto an Evosep. A 44 min gradient was used with an EV1109 column (8 cm × 150 μm, 1.5 μm). Samples were injected into a Thermo Exploris 480 with FAIMS. A -45 CV was used for all runs with 2200 V source voltage. A data independent acquisition method was utilized, where MS1 scans were generated with a 500–740 m/z window followed by 4 m/z DIA windows with optimal window placement. For MS1 a resolution of 30,000, AGC of 300%, and 20 ms max IT were used. For all MS2 scans stepped collision energy of 22, 26, and 30 was used with a resolution of 30,000 and m/z range of 200–1800. An AGC target of 300% and Auto injection times were used. RAW data files were analyzed with DIA-NN 1.8 using a library free deep learning approach. Spectral libraries were generated from an Uniprot human FASTA file with mostly default settings. Precursor charge was set to 2–6 and precursor m/z was 500–740. MBR was used for all runs. Enrichment analysis was performed comparing Resistant and Sensitive tumors using GSEA 4.2.2 following conversion of proteins into gene names.

## Drug compounds

Palbociclib, abemaciclib, ruxolitinib, fedratinib, pacritinib, and itacitinib were purchased from Selleck Chemicals (Houston, TX). Momelotinib was purchased from ChemieTek (Indianapolis, IN).

## Cell culture

MCF7 (ATCC, HTB-22) cells were cultured in RPMI 1640 (Thermo-Fisher, MA) supplemented with 10% fetal bovine serum, 1% penicillin/streptomycin, and 10% MEM non-essential amino acids solution. Cell lines were maintained at 37 °C and 5% $CO_2$, and regularly tested for mycoplasma. Resistant MCF7 Palbo-R and Abema-R cell lines were generated through prolonged culturing of MCF7 cells in palbociclib or

abemaciclib through dose-escalation treatment (10 nM to 1–2 μM) over an 8-month period.

## Immunoblotting

Cell pellets were collected and lysed with Laemmli Sample Buffer (BioRad, Hercules, CA) with beta-mercaptoethanol and heated at 95 °C. Samples were processed with NuPAGE Novex Gel Electrophoresis Systems (ThermoFisher) followed by standard immunoblotting procedure. Immunoblotting antibodies and dilutions are provided in Supplementary Table 2. Secondary antibodies were utilized at a 1:5000 ratio.

## Organoid generation and drug treatment

Organoid models were generated as previously described[79]. In brief, human tumor samples were implanted into NSG mice and collected, and grown in Matrigel (Corning, NY) with organoid media. For drug treatments, organoids were dissociated and seeded into 384-well plates, incubated for 1 day, and treated with serial dilutions of palbociclib or pacritinib for 5 days. Cell viability was measured with Cell Titer Glo 3D (Promega, WI) and luminescence was measured using a Tecan infinite M200 plate reader.

## Cell viability assays and synergy calculations

MCF7 cells were treated with the indicated doses and duration of inhibitors. Following, cell viability was measured with CellTiter-Blue (Promega) and fluorescence assessed with a plate reader. Synergy indices between CDK4/6 inhibitors and pacritinib were calculated using SynergyFinder (synergyfinder.org) following website interface[80].

## IDO1 immunohistochemistry (IHC) and quantification

Five-micron FFPE tissue sections were stained using the Leica BOND automated staining system according to manufacturer's instructions, as previously described[81]. Rabbit anti-IDO1 (Cell Signaling, #86630) was used at a 1:300 dilution with epitope retrieval solution 2. IDO1 IHC slides were scored by a pathologist and quantified using the Allred score based on the proportion and intensity of tumor cells stained. IDO1 positivity was defined as at least 1% staining.

## IDO1 expression and clinical data across additional cohorts

*IDO1* mRNA expression and clinical data from ER+/HER2− BCs were extracted from TCGA[17] and METABRIC[82] datasets available on cBioPortal.

## NeoPalAna resistance gene signature construction and validation

Gene expression data from the NeoPalAna ET-R and NeoPalAna Initial cohorts were analyzed to identify 33 genes combined to distinguishing between Sensitive and Resistant patient groups. Gene expression data from all patients available from NeoPalAna ET-R ($n = 13$ Sensitive, 7 Resistant) and NeoPalAna Initial ($n = 26$ Sensitive, 5 Resistant) cohorts were included. From these two training cohorts, candidate genes were first selected based on fulfilment at least one of the following criteria: (1) Differentially expressed genes (DEGs) between Resistant and Sensitive groups from the NeoPalAna ET-R cohort (915 DEGs from NeoPalAna ET-R identified using DEseq2 with adjusted *p*-value < 0.05, of which 332 were shared with NeoPalAna Initial cohort identified using Limma with a *p*-value < 0.05 given the small sample size of Resistant samples); (2) annotation in key enriched molecular pathways identified through GSEA with FDR *p*-value < 0.05 including E2F targets, G2M checkpoint, Interferon alpha response, Interferon gamma response, MTORC1 signaling, MYC targets V1, and MYC targets V2, in addition to Hallmark Estrogen response early, Hallmark Estrogen response late, and KEGG cell cycle; (3) genes of interest including *MTOR* (not annotated in the Hallmark MTORC1 signaling gene set), key breast cancer markers (*ERBB2* and *ESR1*), and those mediating immunological

pathways (Tumor inflammation signature and HLA genes). Logistic regression models and ROC curve analysis and AUC calculations were performed to assess the predictive value of individual genes, followed by the construction of a multi-gene signature which encompassed representative candidates (Supplementary Table 8). Bootstrapping[30] analysis was performed to identify potential bias and to evaluate optimism correction for AUC (Additional description in Supplementary Information).

In the "Real world" metastatic cohort[31], patients were stratified into "low", "medium", or "high" score groups based on the gene expression score derived from the HTG EdgeSeq Oncology Biomarker Panel ($n = 2560$ genes), of which 27 of the 33 predefined genes from our panel were available for analysis. The low scoring group was utilized as reference group for survival outcome comparisons using logrank tests. Figures were plotted using the survival (v3.6-4), survminer (v0.4.9) packages. Signature construction and reporting were performed considering TRIPOD-AI recommendation[83] (Supplementary Data 2).

## Statistical analysis

The rates of CCCA (overall and by subgroups) were defined as the percentage of patients with tumor Ki67 ≤ 2.7% at C1D15. The corresponding 90% confidence intervals (CI) were calculated using normal approximation or binomial exact CI as appropriate. A two-sample *t*-test was used for age, Wilcoxon rank sum test for Ki67, and Fisher's exact test for categorical characteristics, with all tests being two-tailed.

## Reporting summary

Further information on research design is available in the Nature Portfolio Reporting Summary linked to this article.

## Data availability

RNAseq and WES data generated in this study have been deposited in the Genome Sequencing Archive (GSA), under the accession code HRA009522. Access is restricted to researchers and requires approval by the NGDC Data Access Committee based on a request consistent with participant consent, institutional ethics approval, and data use restrictions, including prohibition of re-identification. Access decisions are typically made within 2–4 weeks, access is granted for a defined research purpose and duration, and the data will remain available under controlled access in accordance with participant consent and applicable regulations. Mass spectrometry proteomic data generated in this study have been deposited in the ProteomeXchange via the PRIDE database, under the accession code PXD058250. Microarray data utilized in this study from NeoPalAna initial cohort under the accession code GSE93204. Transcriptomic data from the Real-world external cohort were from source publication[31]. De-identified individual participant data underlying the results reported in this article will be shared with qualified researchers upon request, subject to institutional approval, a data use agreement, and compliance with patient consent and privacy regulations. Source data are provided with this paper.

## Code availability

*R* code utilized to develop the gene resistance signature is provided in the Supplementary Information.

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

## Acknowledgements

The authors thank the patients and families who participated in this study and the staff who cared for these patients. We thank Stephanie Myles for assistance in protocol development, the Siteman Cancer Center Tissue Procurement Core, Washington University Genomic Technology Access Center (GTAC), and Washington University Anatomical and Molecular Pathology Laboratory. Pfizer provided the study drug palbociclib. This work is funded in part by Siteman Cancer Center Grant (P30 CA91842, SCC, Eberlein), Barnes and Jewish and Siteman Cancer Center Investment Grant (Ma), NIH 1R01CA275904 (Ma and Zhu), NIH CA262804 (Weber), Susan G. Komen Leadership Award (Ma), Breast Cancer Research Foundation (Ma), Saint Louis Men's Group Against Cancer (Ma), David D. Farrell and Helene Urvoaz Farrell Research Fund (Ma), and Pfizer Pharmaceuticals.

## Author contributions

J.L., F.G., and C.X.M. contributed to the conception and study design. C.X.M. and W.Z. contributed to obtaining financial support. S.T. and L.N. provided administrative support. M.O., L.P., F.A., J.M., R.A., K.G., D.N., M.G., and C.X.M. contributed to provision of study materials or patients. T.K., A.M., J.L., M.H., J.H., Z.G., S.T., F.G., L.N., Y.T., J.W., A.K.W., E.S.K. contributed to the collection and assembly of data. T.K., A.M., J.L., M.H., A.Z.W., J.H., Z.G., A.G., S.T., Y.S., F.G., Y.T., S.S., I.H., F.G., M.H., L.D., S.T.O., J.W., A.J.W., E.S.K., R.B., J.D.W., C.X.M., contributed to data analysis and interpretation. T.K., J.L., and C.X.M. wrote the manuscript with critical input from all the authors. All authors contributed to the data interpretation and final approval of the manuscript.

## Competing interests

C.X.M. received research funding from Pfizer Pharmaceuticals. C.X.M. received advisory/consulting fees from Eli Lilly, Stemline, Novartis, TerSera Therapeutics, AstraZeneca, Olaris, Pfizer, Stemline, Daiichi, Merck, Regor Therapeutics, Danatlas. I.S.H. received advisory board fees from AstraZeneca. The remaining authors declare no competing interests.

## Additional information

[1]Divison of Oncology, Department of Medicine, Washington University School of Medicine, St. Louis, MO, USA. [2]Cancer Biology Graduate Program, Division of Biology and Biomedical Sciences, Washington University School of Medicine, St. Louis, MO, Department of Medicine, St. Louis, MO, USA. [3]Divison of Hematology, Department of Medicine, Washington University School of Medicine, St. Louis, MO, USA. [4]Department of Medicine, Weill Cornell Medicine, New York, NY, USA. [5]Department of Neurosurgery, Massachusetts General Hospital, Harvard Medical School, Boston, MA, USA. [6]Division of Public Health Science, Department of Surgery, Washington University School of Medicine, St. Louis, MO, USA. [7]Section of Endocrine and Oncologic Surgery, Department of Surgery, Washington University School of Medicine, St. Louis, MO, USA. [8]Department of Pathology and Immunology, Washington University School of Medicine, St. Louis, MO, USA. [9]Baylor College of Medicine, Houston, TX, USA. [10]Department of Biochemistry and Molecular Medicine, George Washington University School of Medicine and Health Sciences, Washington, DC, USA. [11]Bursky Center for Human Immunology & Immunotherapy, Washington University School of Medicine, St. Louis, MO, USA. [12]Department of Molecular and Cellular Biology, Roswell Park Comprehensive Cancer Center, Buffalo, NY, USA. [13]Department of Pathology, Roswell Park Comprehensive Cancer Center, Buffalo, NY, USA. [14]Department of Medical Oncology, Mayo Clinic, Rochester, MN, USA. [15]Division of Hematology and Medical Oncology, Mayo Clinic, Phoenix, AZ, USA. [16]These authors jointly supervised this work: Jingqin Luo, Cynthia X. Ma. ✉e-mail: jingqinluo@wustl.edu; cynthiaxma@wustl.edu

