## [Transparent Peer Review file · Nature Communications]

Biomarkers of response to neoadjuvant palbociclib plus anastrozole in endocrine-resistant estrogen receptor-positive/HER2-negative breast cancer: a phase 2 trial

Corresponding Author: Dr Cynthia Ma

Version 0:

Reviewer comments:

Reviewer #1

(Remarks to the Author)

Biomarkers of Response to Neoadjuvant Palbociclib Plus Anastrozole in Endocrine-Resistant Breast Cancer

The manuscript investigates potential biomarkers of response to a combination treatment of palbociclib and anastrozole (ANA/PAL) in estrogen receptor-positive (ER+)/HER2-negative breast cancer that is resistant to standard endocrine therapy. In this study, the authors analyze tumor biopsies from 34 patients to identify molecular features associated with treatment resistance or sensitivity. The primary endpoint was the rate of complete cell cycle arrest (CCCA), defined by a significant decrease in the proliferation marker Ki67.

Key findings include:

- A 57.6% CCCA rate was observed among patients receiving ANA/PAL, indicating that this treatment combination may suppress proliferation in a subset of endocrine-resistant tumors.
- Resistance to ANA/PAL was associated with certain intrinsic subtypes (e.g., LumB and HER2-E) and the activation of signaling pathways related to cell cycle progression (e.g., MYC and mTOR), immune response (e.g., interferon), and immune tolerance markers such as IDO1.
- The authors propose that certain molecular markers, particularly gene signatures related to cell cycle control and immune modulation, could serve as predictive biomarkers of response to ANA/PAL.

Major Issues

While this study provides a detailed descriptive analysis of biomarkers in a small cohort of endocrine-resistant breast cancer patients, the following key issues limit its suitability for publication in a high-impact journal like Nature Communications:

1. **Small Sample Size (34 Patients)** The small sample size significantly limits the generalizability and statistical power of the study's findings. With only 34 patients, there is a higher risk of variability and limited ability to draw reliable conclusions about potential biomarkers of resistance or sensitivity to ANA/PAL. For studies of predictive biomarkers in cancer, larger cohorts are typically needed to achieve statistical robustness and validate associations between molecular characteristics and treatment outcomes.
2. **Descriptive Nature Without Experimental Validation** The study is largely descriptive, identifying potential biomarkers without functional experiments to validate their roles in modulating response to ANA/PAL. It is just a number of -omic studies conducted in a sampleset with a clinical outcome, but not real hypothesis-driven scientific work. For a biomarker study intended for a journal like Nature Comms experimental validation is essential. This could involve in vitro or in vivo assays to confirm whether the identified markers directly influence treatment efficacy. Without such experiments, the study does not establish causative relationships, making it less impactful for the field.
3. **Lack of an Independent Validation Cohort** An independent validation set is absent in this study, which limits the reproducibility and reliability of the findings (kind of in line with point 1 – but equally essential). High-impact studies in biomarker research typically require validation in a separate cohort to confirm that observed associations are not specific to the initial sample. The absence of a validation cohort reduces confidence that these biomarkers will be effective in broader patient populations.
4. **Redundancy of Certain Findings** Some findings, such as the association of intrinsic subtype with response to CDK4/6

inhibitors, are already well-documented in the literature. The association between high-risk subtypes like LumB and reduced responsiveness to CDK4/6 inhibitors has been previously established (e.g., Aleix Prat et al, Journal of Clinical Oncology, 2021; 39(13): 1458-67), which diminishes the novelty of this manuscript. While the study provides additional detail on molecular signatures, the insights presented do not substantially advance current knowledge.

Conclusion

In summary, while the manuscript offers an informative descriptive analysis of potential biomarkers in endocrine-resistant breast cancer, these limitations in sample size, experimental work, lack of independent validation, and the redundancy of some findings preclude its publication in a high-impact journal like Nature Communications. Strengthening the study with a larger cohort, experimental validation of biomarkers, and a validation set would significantly enhance its impact and suitability for such a journal.

Reviewer #2

(Remarks to the Author)

Biomarkers of response to neoadjuvant palbociclib plus anastrozole in endocrine-resistant estrogen 1 receptor-positive/HER2-negative breast cancer

Kong et, al described a translational study based on the data collected from the NeoPalAna trial in which stage II/III ER+/HER2- BC patients were treated with ANA/PAL. Responding patients were compared with the resistant ones, on multiple -omics levels, with the purpose to understand the mechanism of resistance and to seek for response biomarkers. The study highlighted several dysregulated oncological pathways, different immune microenvironment, and protein IDO1 which is associated with PAM50 subtype, in patients who did not respond to ANA/PAL.

The paper is well presented, and the data is valuable for audience in oncology research community. The conclusion made so far has been properly underpinned by the analysis results. However, the authors have primarily focused on innate resistance mechanism while barely touched response biomarkers, as opposed to what its title suggested. The novelty of the conclusions is limited. The data could be further analysed to deliver its full potential.

Main points:

1. The study has a classical design: patients were categorised into responding and resistant cohorts, while samples were taken at pre-treatment and post-treatment, collectively forming a two-by-two data structure. Throughout comparisons have been made between the responding and resistant cohorts, at each individual time point, while little was compared between the pre- and post- treatment, within each patient cohort. The authors should characterise the changes at transcriptomics and proteomics level following ANA/PAL, and define the CHANGES that differentiate the responding vs. resistant patients.
2. The authors have rightly pointed out that using ki-67 as a surrogate for treatment response has its limitation. While acceptable in clinical trial design, this have a more prominent impact in biomarker discovery. In this study ki-67 acted as a perfect biomarker for response, leaving no chance a novel treatment-specific biomarker to be discovered. The authors should introduce objective response and other clinical endpoints for response biomarker discovery, followed with exploration on the relationship between the identified biomarkers and ki-67.
3. Genomic, transcriptomic, proteomic, and cellular data were present separately without any attempt to provide an integrated view.

Minor points:

1. The authors have associated -omics changes with ki-67 (267-290), but only provide description on a handful of selected gene/proteins. A better overview at -omics level, maybe in the format of correlation network topology and its changes, will help the audience understand the systematic omics-landscape shifting related to cell-proliferation as a result of ANA/PAL treatment.
2. The number of DEGs are large. Maybe the limited sample size on the homogenous responding patients played a role? PCA showed heterogeneous omics profile in different BC subtypes. Is there a subtype-specific pattern in the DEG?
3. ANA/PAL-resistant and ANA/PAL-R have been used in a mixed way (234-235, and more).

Reviewer #3

(Remarks to the Author)

In the manuscript reviewed, Kong, et al. utilizes the NeoPalAna endocrine-resistant cohort to explore the potential benefits of adding a CDK4/6 inhibitor to endocrine therapy (ET). Their findings demonstrate that combining palbociclib (PAL) with anastrozole (ANA) effectively suppresses cell proliferation resistant to ANA monotherapy, achieving complete cell cycle arrest (CCCA) in 57.6% of ET-resistant ER+/HER2- breast cancer (BC) cases by cycle 1, day 15 of treatment (C1D15). They then investigated potential mechanisms of resistance to the ANA/PAL combination, through a detailed genomics analysis of sensitive and resistant tumors. While no major genetic differences were identified, ANA/PAL-resistant tumors exhibited upregulation of cell cycle genes, suppression of G2M checkpoint genes, and activation of MYC and mTOR signaling pathways—consistent with continued proliferation (Ki-67 positivity). Most interesting is that the PAM50 classification was able to separate responders and non-responders. Additionally, some differences were observed in immune cell fraction induction (via CIBERSORT) such as presence of higher T-cell-inflamed TME and IDO1 levels in resistant tumors which could warrant the use of immune checkpoint inhibitors. Overall, despite the limited cohort size, this dataset represents a

valuable, well-characterized resource for advancing research in ET resistance and potential therapeutic strategies in ER+/HER2- BC. Moreover, the results provide significant starting point for a biomarker panel that can identify responders to ANA/PAL and those who will need other treatments. Importantly, the genomic data suggests alternative paths for these resistant individuals.

Questions for authors:

1. The primary findings of significant clinical interest are: that 1) at baseline (BL), luminal A tumors are uniformly sensitive whereas HER2+/Basal were insensitive with Luminal B tumors being split in sensitivity and resistance; 2) higher BL Ki67 is clearly associated with resistance, and 3) evidence for immune activation in resistant tumors at baseline. However, simple clinical characteristics can also discern responders and non-responders. It would be helpful to address whether the molecular characteristics (e.g. taking the best biomarkers for response, like Ki67) would improve the prediction of response.
2. There were issues that the authors do not comment on such as the relevance of T stage and more interestingly, race being significantly associated with response (Table 1). The observations are limited by the cohort size but how does this compare with other datasets? For example, do white patients respond better and Asian patients do not? Importantly, all patients underwent resection after the cycles of ANA/PAL and I assume that Table 2 represents these data, but surprisingly, there was no attempt to address whether any of the biomarkers were correlated with this important endpoint. This seems like a major flaw that is easily amenable in a revision. For example, there were PAM50 shifts at C1D15. Did any of those with shifts correlate with CR or PR?
3. While these BL observations are noteworthy and potentially clinically important, the impact of the C1D15 seems less clear but has significant biological interest. That there were such significant expression changes only 15 days after initiation of ANA/PAL treatment is intriguing and is not likely to be related to clone fluxes rather to direct transcriptional effects of the therapy. Yet, other than a descriptive narrative, the authors seemed to have avoided much conjecture as to the potential biology, nor did they integrate these data into the final narrative. Moreover, Figure 4, which depicts the C1D15 data is poorly annotated so it is not clear to this reviewer how to interpret some of the data. For example, the labels for the X and Y axis in panel G do not explain what sensitive r means and what is the interpretation of why CCND1 and AKT1 is below the dotted line. This all seems like a wasted opportunity. The shift in PAM. That the immune markers for IFN signalling (Figure 4E) actually increased in C1D15 is of interest, perhaps the authors could, in the discussion, explore this a bit further. This seems to be contrary to the cibersort results in Figure 6 where the C1D15 levels of the different immune subtypes do not appear to be going up. Again, this may be the confusion of what each of the figures are referring to – BL, or C1D15, For the patient tumors that were classified as ANA/PAL sensitive at C1D15, it appears that most of them maintained their relative sensitivity based on clinical response reported in Table 2. What were the three tumors that had stable disease? Did they have any specific characteristics (PAM50, grade, genetic or transcriptional signals etc) compared to the tumors that had partial or complete response?
4. Similarly, since PAM50 subtype switching is observed in the tumors before and after treatment, it would be simpler to visualize the data in a manner (plot or table) that shows the subtypes before and after treatment for each tumor instead of just a Venn diagram showing the subtypes pre and post treatment in Fig. 3A and 4A. Is there any association with PAM50 subtyping and response at C1D15? Also Table 1 has “PAM50” notes as “PAM5”.
5. Since WES data was generated, have the authors looked for mutation calls across of genes vs just the 83 genes reported here (Fig.2)? Were any other highly occurring mutations observed in the cohort? Were any of them associated with resistance or sensitivity? What are the criteria for selection of genes shown in copy number heatmap in Fig.2B? All the genomic analysis does not appear to be comprehensive enough to conclude that there is no mutation of copy number changes associated with response.
6. The limited size of the dataset was probably why no significance could be established on the relevance of the PIK3CA mutation incidence in ANA/PAL-resistant vs sensitive cases. Are there any other publicly available datasets that could be analyzed for this? The authors referred to other similar studies with biomarkers but there did not seem to be a systematic comparison of the biomarkers across these studies.
7. In Fig.2A, the authors should describe what the S and R barplots on the right side are. Is it the percentage of all samples with the mutation in each category? The barplot in the top panel of the figure also has no y axis label. In Fig.2B please add in legend for sensitive/resistance sample annotation at the side of the heatmap.
8. “Top positively enriched DEGs..” described in line 235 don’t really all appear to be “top” DEGs in the volcano plot in Fig.3D by fold change or fdr. Please describe the “top” DEGs and rephrase the rationale for highlighting the genes listed.
9. In Figure 3E and 4E, please include fdr or p value for the pathway enrichment analysis. Only NES is shown. The text (lines 236-240) reports the pathway enrichment analysis but don’t really highlight the relevance of this analysis. Please rephrase text accordingly.
10. In PCA plots in Fig.3C and 4C, the basal subtype tumors always separate out from all other tumors (irrespective of being sensitive or resistant) on PC1. Would you see a separation of the resistant and sensitive tumors if you remove the three basal tumors from the analysis?
11. In Fig.5D and 7A please mention what the change in the circle size represent.
12. In fig.6B, despite seeing a higher level of CD8+ T cells and M1 macrophages at baseline, the tumors still acquire resistance to ANA-PAL treatment. Moreover, at C1D15 there is no significant difference in the levels of T cells, NK cells, M1 and M2 macrophages between sensitive and resistant tumors. The interpretation of these results can be commented on more in the discussion. For example, looking at absolute levels of immune cells post treatment may not be indicative of response in these tumors?
13. The first p-value line in Suppl. Fig S2A, appear to be marked wrong. Are the authors comparing NeoPalAna ET-resistant cohort (pink) compared to that of the Z1031_POL ET-sensitive cohort (yellow)?

Reviewer #4

(Remarks to the Author)

Version 1:

Reviewer comments:

Reviewer #1

(Remarks to the Author)

The new data and experiments improve the overall quality of the manuscript, addressing the majority of the concerns raised by this reviewer. Although the number of patients cannot be changed, the validations and the experiments make the whole work more convincing and suitable for this journal.

Reviewer #2

(Remarks to the Author)

Substantial changes have been made to the manuscript following the reviewers' comments. With additional datasets and new analysis, the authors were able to characterise -omics landscape related to cell proliferation and changes following CDK4/6i. The comparison between resistant and response samples, at molecular levels and functional levels, provide insight to the mechanism underpinning treatment response. The new 33 gene baseline model is a good add-on.

The sample size unfortunately limited the depth of this research, nonetheless I am satisfied with the level of evidence underpinning the main conclusion of the paper, and found that the major concerns from the reviewers have been addressed to a satisfactory level.

The only addition I would like to suggest is to include in this manuscript is a summary table of RNA/protein data related to the 33 gene (BL predictors) in the supplementary material.

Reviewer #3

(Remarks to the Author)

On the whole, this revision of the work is improved. Integration of RNA-seq and mass spectrometry analysis and Transcriptomic changes between C1D15 and baseline in Resistant versus Sensitive tumors improve the paper significantly. Moreover, the addition of the validation cohort was very helpful.

However, there are several issues that need some attention:

The additional analysis with the 33 genes signature on the three datasets indeed looks promising and makes good utilization of all the data generated. The authors however, should clearly elaborate on how the 33 genes were derived/nominated based on the NeoPalAna ET-R and NeoPalAna Initial data. The number 33 seems arbitrary and without any rationale. Were they picked because they were overlapping genes in both the dataset sensitive vs resistant comparisons or had molecular pathway relevance etc? In Fig.9B could you highlight the genes based on the different categories described (cell cycle, oncogenic signaling, proliferation, and immune and interferon response mediators etc). This is a major oversight.

The discussion around figure 6 is rather confusing. While the expression analysis and western blot data on the MCF7 in vitro analysis looks promising the in vitro drug response data is not convincing. For example - Authors justify the use of JAK/STAT inhibitors based on the expression analysis on Palbo-R vs parental and Abema-R MCF7 vs parental cells. However there is no real difference (increased sensitivity) observed between parental and Palbo-R or Abema-R cell lines for any drug or dose (Fig.6H). In fact, it seems that the resistant cell lines are perhaps less sensitive than the sensitive line. The data just shows that all the 3 cell lines have increased cell death for increasing concentrations of most drugs - which would be expected? If the data does not show that Palbo-R or Abema-R MCF7 cells have increased sensitivity to JAK inhibitors compared to parental cells what are the authors trying to state with the data?

"Screening with these inhibitors and the JAK1 selective inhibitor itacitinib in parental MCF7 and CDK4/6i-resistant cells suggested a higher dependency on JAK2 than JAK1, with similar sensitivity across the cell lines (Fig. 6H)."

That Pacritinib is more effective on higher doses tested (in all three cell lines) compared to other drugs could just be a function of concentrations tested in the experiment. That no cell death is observed in response to Itacitinib could just be inadequate concentrations tested in this experiment. In order to draw this conclusion of increased dependency on JAK2 vs JAK1 authors will have to normalize to the JAK molecule inhibitory effect of the drugs for a given concentration or perform gene knock out experiments to confirm the JAK1 vs JAK2 (and IRAK1) sensitivity.

For the in vitro drug data on organoids derived from PDX tumors (Fig.6K and 6L), the authors state that "Both organoid models showed reduced sensitivity to palbociclib (Fig. 9K) but were responsive to pacritinib at low micromolar doses (Fig. 9L), which was comparable to the response observed in MCF7 Palbo-R/Abema-R cells. These data support the potential

importance of JAK/STAT signaling in CDK4/6i resistant BCs." Not sure if the data shown is enough to compare sensitivities between drugs and make this statement - these are two different drugs with two completely different mechanisms of action. So if WHIM43 for example has an IC50 of 1.9uM to Pacritinib (JAK2 inhibitor) and 3uM to Palbociclib (CDK4/6 infinite) does it have any significant biological implication? Also figures 6K and 6L have been referred to wrongly in the text.

"Zero Interaction Potency (ZIP) drug synergy calculations of CDK4/6i and pacritinib at indicated doses." What are the doses? There is no description of how this synergistic experiment was performed and how should one interpret the colors etc. The legend was not helpful.

Authors should describe in results or legend data in figured better - for example in fig 6G -western blot - There is no mention of AKT and RB results.

Reviewer #4

(Remarks to the Author)

Reviewer #5

(Remarks to the Author)

As a statistical reviewer, I will focus this review on the methodology of the clinical trial and the development of the molecular signature.

To summarize this work, the authors studied the mechanisms of resistance in ER+ breast cancer to the combination of anastrozole + palbociclib. Translational experiments are performed on samples of patients included in a single-arm phase II Simon's two-stage study. In this study, the combination is used in a neoadjuvant setting, for patients with large tumors (T2 or more) whatever the lymph node status and without metastatic disease.

The clinical study was stopped before the end of recruitment after the decision of the data and safety monitoring board for efficacy, which is an unusual situation, as Simon two-stage design allow to stop the study for futility, not for efficacy.

The primary endpoint is the rate of complete cell cycle arrest (CCCA) after 1 month of combination of anastrozole + palbociclib, assessed by Ki67% immunohistochemistry and centrally reviewed. The sample size is reproducible, though the hypotheses are very pessimistic (unacceptable rate of 5%, acceptable rate of 20% of CCCA).

In the analysis of the primary endpoint, there were 22/33 (66%) of patients that exhibited CCCA at C1D15.

Major comments:

- My main concern on that work is about the 33 genes signature that would be predictive of resistance to the combination of anastrozole + palbociclib (ANA+PAL). There is first a strong issue of reporting. The authors can look at the TRIPOD recommendations, that have been recently updated for AI-based prediction models (PMID: 38626948). Even though these recommendations are for dedicated work on prediction models (not integrated to a translational work), it could serve as a basis for the reporting of this part of the work. There is no info on the selection process of the variables (and their possible transformation) into the model, nor the nature of the model used to estimate the probability of being resistant (logistic regression model? Others models?).

- Then, probably the biggest issue, there is only 33 patients to develop this model (with 11 patients that presented the endpoint "resistance to ANA+PAL") which is very problematic if genes are introduced as unique variables in the analysis (33 potential variables). For example, a very loose rule of thumb is 10 events for 1 variable in a logistic regression model (therefore in the case of 11 events, only 1 variable can explain the outcome). And finally, there is no correction of optimism when estimating the performances of the gene signature (e.g. AUC) by any technique (for example, bootstrap, see PMID: 38191193). The optimism is linked to the fact that the same data are used to choose the model and evaluate its performances.

- Regarding the external validation of the score, there is several issues. First, the score has been developed to predict the ANA+PAL resistance (binary, yes or no) in the neoadjuvant setting, and is validated on another outcome (progression-free survival). This is therefore not a good external validation (for more information, see PMID: 38224968). Second, there is no information on how are chosen the thresholds "low", "medium" or "high" for expression of the signature (quartile? distributions?). On the Kaplan-Meier curves, the 95% confidence intervals are not displayed, but when looking at the number of patients at risk, we can see that the subgroups are quite small.

- Regarding clinical response, only 21/34 patients that completed the 5 cycles had a reported clinical response. This means that this response rate is evaluated only in sensitive patients and not in resistant patients that stopped the treatment. This should be more transparent in the narrative. In addition, in an intention-to-treat analysis, which is recommended to avoid to inflate the effect of the treatment, the denominator of the response rate should be 34 and not only 21 as current.

- It is written that it was a RECIST evaluation (radiological evaluation based on target lesions), but the rhythm of assessment is not defined. It is a response at a given time point (e.g. at surgery) ? Is it the best overall response on treatment? This point deserves some clarification.

Minor comments:

- The authors stated in the responses to the rebuttal letter that "As discussed earlier in response to previous reviewers' comments, clinical characteristics were insufficient to discern responders vs non-responders (Table 1)", which is not completely true, as tumor size (T3 versus T2) and grade (III versus I-II) were associated with resistance, and may have been

included in the model that predicts resistance to ANA+PAL

- The CCCA seems to be used more and more in the neoadjuvant setting for breast cancer as an endpoint, but its relevance for patients could be debated. Notably, the rate of patients that managed to undergo surgery is probably a more meaningful clinical endpoint, even though it is understandable in the setting of this translational study that this endpoint has been chosen

- The one-sided error rate is 10% for the computation of the sample size, therefore if the authors want to compute confidence intervals that mirror the sample size, it is not a 90% coverage but an 80% coverage that should be chosen. Anyways, as this is not a sample size based on confidence interval, it is not mandatory to reduce the coverage of confidence intervals

To conclude, from the statistical perspective, the paper does not meet the standards of a well-conducted clinical research study. I would expect that Nature Communications would require a better methodology for clinical research, as they do for basic science.

Version 2:

Reviewer comments:

Reviewer #3

(Remarks to the Author)

I have reviewed this revision of the manuscript and believe it is significantly improved. Though the numbers are small, the use of a validation cohort, and the underlying biology uncovered is sufficiently interesting and should direct the investigations in larger cohorts.

(Remarks on code availability)

Reviewer #4

(Remarks to the Author)

(Remarks on code availability)

Reviewer #5

(Remarks to the Author)

Statistical review.

Regarding my main concern—the development of the signature—the authors have significantly improved the transparency of the reporting of their results. However, several issues remain:

- Could you please annotate the code regarding correction for optimism of AUC ? I am not sure to understand what have been done.

See 10.1136/bmj-2023-074819 “Bootstrapping is a resampling technique, where a bootstrap sample is created by randomly sampling (with replacement) from the original data. In the enhanced bootstrap, a model is developed (repeating the model building steps used to develop the model on all the data) in each bootstrap sample and its performance evaluated in this sample as well as the original dataset to get an estimate of optimism of model performance. This process is repeated many times and the average optimism calculated, which is then subtracted from the apparent performance.”

- Why does the methods section not mention that the signature is defined as the averaged Z-score of the genes, as stated by the authors in their rebuttal letter?

- Why was bootstrap validation performed on each gene independently, rather than on the composite model using the signature and/or Ki67 as the dependent covariate?

The manuscript would also benefit from clarifying some points:

- Line 213, page 9: The phrase “in an unbiased manner” should be removed. The authors incorporated expert knowledge in the selection of covariates, which, while acceptable in model development, cannot be considered unbiased.

- Line 226, page 9: The term “iterative selection” is unclear. The authors should clarify what this process entailed

- Line 459, page 19: The authors should specify the exact sample size of the cohort used to develop the score (i.e., patients with RNA-seq data). Based on the current study, this appears to be $n = 20$, but it is unclear whether additional samples were included.

- Line 568, page 24: The statement “This provided validation...” should be tempered. The results supporting the gene signature are fragile (very small sample sizes for the development and validation cohorts, different outcomes for the development and the validation of the score) and should be presented with appropriate caution. As mentioned by the authors in the rebuttal letter “future large studies are needed to confirm and generalize these findings.”

(Remarks on code availability)

The code is provided, but cannot be run without the source file, which is understandable given that it contains patient data. To enhance clarity, additional annotations would be helpful, in particular for the procedure of correction of AUC for optimism (see previous comment).

Response to Reviewers' Comments

We thank the reviewers for the thoughtful comments for our manuscript titled: "Biomarkers of Response to Neoadjuvant Palbociclib Plus Anastrozole in Endocrine-Resistant Breast Cancer". We have read all the comments in detail and have addressed all of the concerns. Below, we provide a summary of the major additions to the manuscript as well as a point-by-point response to each of the reviewers' comments. We hope that our revisions and responses meet with your approval.

Our major additions to the manuscript are summarized below:

1. We performed additional analyses of the proteogenomic data, including:
 - a. Transcriptomic changes between C1D15 and baseline in Resistant versus Sensitive tumors (New Fig. 5).
 - b. Integration of RNA-seq and mass spectrometry analysis for C1D15 samples (New Fig. 4).
 - c. Validation of differentially expressed genes and pathways in Resistant versus Sensitive tumors at baseline using gene expression data from the NeoPalAna Initial cohort (New Suppl. Fig. 4).
2. We performed in vitro validation of JAK/STAT signaling, which is downstream of IFN and inflammatory pathways, in resistant ER+ breast cancer cells:
 - a. We analyzed differentially expressed genes and pathways in MCF7 cells with acquired resistance to CDK4/6 inhibitors through chronic treatment with palbociclib or abemaciclib (Fig. 6).
 - b. We examined the anti-tumor activity of a series of clinical-grade JAK1/2 inhibitors in CDK4/6 inhibitor resistant MCF7 cell lines and ER positive patient-derived xenograft (PDX) organoid models (New Fig. 6).
3. We derived a CDK4/6i-resistance 33-gene signature using data from the NeoPalAna ET-R cohort and the data from the NeoPalAna initial cohort, and validated it in an independent external cohort below:
 - a. Metastatic HR+/HER2- cohort (NCT04526587), treated with CDK4/6i in combination an aromatase inhibitor (AI) or fulvestrant (FUL): 151 patients (115 AI + CDK4/6i; 36 fulvestrant + CDK4/6i) (New Fig 9).

We believe that these findings have strengthened the robustness of our findings.

Please find our point-by-point responses to each of the reviewers' comments below:

Reviewer #1 (Remarks to the Author):

Biomarkers of Response to Neoadjuvant Palbociclib Plus Anastrozole in Endocrine-Resistant Breast Cancer

The manuscript investigates potential biomarkers of response to a combination treatment of palbociclib and anastrozole (ANA/PAL) in estrogen receptor-positive (ER+)/HER2-negative breast cancer that is resistant to standard endocrine therapy. In this study, the authors analyze tumor biopsies from 34 patients to identify molecular features associated with treatment resistance or sensitivity. The primary endpoint was the rate of complete cell cycle arrest (CCCA), defined by a significant decrease in the proliferation marker Ki67.

Key findings include:

-A 57.6% CCCA rate was observed among patients receiving ANA/PAL, indicating that this treatment combination may suppress proliferation in a subset of endocrine-resistant tumors.

-Resistance to ANA/PAL was associated with certain intrinsic subtypes (e.g., LumB and HER2-E) and the activation of signaling pathways related to cell cycle progression (e.g., MYC and mTOR), immune response (e.g., interferon), and immune tolerance markers such as IDO1.

-The authors propose that certain molecular markers, particularly gene signatures related to cell cycle control and immune modulation, could serve as predictive biomarkers of response to ANA/PAL.

Major Issues

While this study provides a detailed descriptive analysis of biomarkers in a small cohort of endocrine-resistant breast cancer patients, the following key issues limit its suitability for publication in a high-impact journal like

Nature Communications:

1. Small Sample Size (34 Patients) The small sample size significantly limits the generalizability and statistical power of the study's findings. With only 34 patients, there is a higher risk of variability and limited ability to draw reliable conclusions about potential biomarkers of resistance or sensitivity to ANA/PAL. For studies of predictive biomarkers in cancer, larger cohorts are typically needed to achieve statistical robustness and validate associations between molecular characteristics and treatment outcomes.

We agree with the reviewer that our sample size was relatively limited. As described in our study design, study enrollment completed after 34 patients as the primary endpoint had already been met with the smaller sample size, per the data safety and monitoring board.

Based on the altered proteogenomic findings, we derived a baseline gene signature predicting response to palbociclib + endocrine therapy. We utilized both the 1) NeoPalAna ET-R cohort and 2) our previously published NeoPalAna initial cohort (NeoPalAna Initial) to increase our sample size. Microarray gene expression data from pre-treatment tumors was available for 31 patients (26 Sensitive and 5 Resistant) enrolled in NeoPalAna Initial (PMID 28270497).

We first assessed for similarity between the Resistant tumors in the NeoPalAna ET-R cohort and the Initial cohort. DEGs of interest from the NeoPalAna ET-R (*BRCA2*, *EZH2*, *IDO1*, *CCNE1*, *IRF4*, *CD48*, *CD38*; Fig. 3D), were also found to be significantly upregulated or near significantly upregulated in association with Resistance to ANA/PAL in NeoPalAna Initial (New Supplementary Fig. 4A, and below). Unbiased GSEA of Resistant vs Sensitive patients in the NeoPalAna Initial also identified upregulation of shared pathways including E2F targets, G2M Checkpoint, Interferon gamma response, Interferon alpha response, IL6-JAK-STAT, mTORC1 signaling, and MYC targets; relevant downregulated pathways included Estrogen response early (New Supplementary Fig. 4B, and below). These data indicate that two NeoPalAna cohorts share similar biomarkers of response to ANA/PAL.

Using the NeoPalANA ET-R and Initial cohort for discovery/training, we nominated 33 genes, among which incorporated key regulators of the cell cycle, oncogenic signaling, cellular proliferation, and mediators of immune and interferon response as part of our NeoPalAna resistance signature (New Fig. 9A-D). We generated Receiving Operating Characteristic (ROC) curves and compared the performance of our gene signature to that of the baseline Ki67 by IHC. In NeoPalAna ET-R, the Area Under Curve (AUC) for Ki67 was 0.824 (95% CI 0.637,1) while the AUC for the NeoPalAna resistance signature was 0.967 (95% CI 0.902,1) (New Fig. 9E).

Adding Ki67 to our gene signature did not improve the performance, with an AUC of 0.967 (95% CI 0.902,1). A greater AUC suggests a higher ability for test discrimination, with conventional standards 0.7-0.8 being considered acceptable, 0.8-0.9 as excellent, and >0.9 as outstanding (PMID 20736804). In NeoPalAna Initial, the AUC of Ki67 was 0.6 (95% CI 0.372, 0.828) while the AUC of the gene signature was 0.992 (95% CI 0.971,1), and combining the two 1.0 (95% CI 1, 1) (New Fig. 9F). Overall, the NeoPalAna resistant gene signature demonstrated high sensitivity/specificity in distinguishing between Sensitive and Resistant tumors.

For an external validation cohort, we collaborated with Dr. Erik Knudsen's group (Roswell Park) for an independent analysis of their "Real world" cohort (PMID 37704753), where targeted RNA-seq analysis of pre-treatment samples and clinical outcomes were available for 151 metastatic HR+/HER2- breast cancer patients were treated with first-line CDK4/6i (~90% palbociclib) in combination with hormonal therapy (115 with aromatase inhibitors +/- prior endocrine therapy, 36 Fulvestrant). We examined whether our resistant gene signature was prognostic for progression-free survival. Blinded with the clinical outcome data, the WashU group (TK, JL) applied the NeoPalAna resistance gene signature on pre-treatment RNA-sequencing data and assigned with low, medium, or high gene signature scores. An independent statistician (JW) from Dr. Knudsen's group then performed K-M analysis of patients stratified by the gene signature score. A higher resistant gene signature score was significantly associated with shorter PFS (HR = 2, $P = 0.012$) in the overall patient population (New Fig. 9G left panel, and below). Similar poorer outcome with shorter PFS, was seen in patients who received AI plus CDK4/6i (New Fig. 9G middle panel) and a trend in those that received CDK4/6i with fulvestrant (New Fig. 9G right panel), although fewer patients received this regimen. A higher gene signature was also associated with a worse overall survival (New Fig. 9H, HR = 2, $P = 0.03$). The analysis conducted in the NeoPalAna Initial and the independent "Real world" Metastatic cohort provided further validation of resistant biomarkers identified from the NeoPalAna cohorts.

2.Descriptive Nature Without Experimental Validation The study is largely descriptive, identifying potential biomarkers without functional experiments to validate their roles in modulating response to ANA/PAL. It is just a number of -omic studies conducted in a sampleset with a clinical outcome, but not real hypothesis-driven scientific work. For a biomarker study intended for a journal like Nature Comms experimental validation is essential. This could involve in vitro or in vivo assays to confirm whether the identified markers directly influence treatment efficacy. Without such experiments, the study does not establish causative relationships, making it less impactful for the field.

We agree with the reviewer that additional validation would increase the impact of our findings. To address this, we investigated if we could recapitulate resistance mechanisms *in vitro* and identify potential novel therapeutic strategies to overcome resistance. We first generated palbociclib-resistant (Palbo-R) and abemaciclib-resistant (Abema-R) MCF7 cells through prolonged culturing in the presence of CDK4/6i inhibitors through progressive drug titration (New Fig. 6A, B, and below). Subsequent RNA-seq of MCF7 Palbo-R and MCF7 Abema-R cells demonstrated shared upregulation of Interferon alpha and gamma signatures, in addition to KRAS signaling, TNF signaling via NFKB, and EMT (New Fig. 6C-F). Indeed, we also observed elevated Interferon signaling in Resistant tumors at both Baseline and C1D15 timepoints compared to Sensitive tumors in the NeoPalAna ET-R cohort. Previous findings have identified similar aberrant interferon signaling linked to intrinsic CDK4/6i resistance mechanisms and pathway activation following drug therapy (PMIDs 33536276, 30867218).

Upregulated IFN/inflammatory signaling was confirmed by increased STAT3 phosphorylation and elevated levels of the downstream interferon-stimulated gene 15 (ISG15) on immunoblot (New Fig. 6G). We then assessed targeting JAK/STAT signaling in CDK4/6i-resistant MCF7 models using JAK inhibitors. JAK inhibitors are clinically approved to treat inflammatory, autoimmune, and neoplastic diseases including graft-versus-host disease (targeting JAK1/2) and myeloproliferative neoplasms (targeting JAK2, including ruxolitinib, fedratinib, momelotinib, and pacritinib). Screening with the various JAK inhibitors with different selectivity against JAK2 and JAK1 across parental MCF7 and CDK4/6i-resistant cells showed increased dependency on JAK2 compared to JAK1 and similar sensitivity between lines (New Fig. 6H). We then assessed pacritinib, as it exhibited the highest anti-cancer potency and targets both JAK-STAT and TNF signaling via NFKB pathways through inhibition of JAK2 and IRAK1, where IRAK1 is a potent transducer of NFKB/inflammatory signaling, and was also found to strongly correlate with Ki67 in Resistant tumors at C1D15 at both RNA and protein levels (New Fig. 4F, G). In Palbo-R cells, pacritinib alone suppressed STAT3 phosphorylation and downregulated cyclin D1 and combination with palbociclib led to additive anti-cancer effects (New Fig. 6I, J). In contrast, the combination of abemaciclib with pacritinib was synergistic (New Fig. 6I, J). We also tested pacritinib in ER+/HER2- patient-derived xenograft (PDX) organoid models (New Fig. 6K, L). Both organoids were resistant to palbociclib but were responsive to pacritinib with IC₅₀ of 1-2 μM, which is achievable clinically. These data support the importance of elevated interferon/JAK-STAT in driving CDK4/6i-resistant tumors, and the potential of JAK2 inhibitors in treating CDK4/6i resistant breast cancer.

3.Lack of an Independent Validation Cohort An independent validation set is absent in this study, which limits the reproducibility and reliability of the findings (kind of in line with point 1 – but equally essential). High-impact studies in biomarker research typically require validation in a separate cohort to confirm that observed associations are not specific to the initial sample. The absence of a validation cohort reduces confidence that these biomarkers will be effective in broader patient populations.

We thank this reviewer for highlighting the importance of having validation cohorts. Please see our response above.

4.Redundancy of Certain Findings Some findings, such as the association of intrinsic subtype with response to CDK4/6 inhibitors, are already well-documented in the literature. The association between high-risk subtypes like LumB and reduced responsiveness to CDK4/6 inhibitors has been previously established (e.g., Aleix Prat et al, Journal of Clinical Oncology, 2021; 39(13): 1458-67), which diminishes the novelty of this manuscript. While the study provides additional detail on molecular signatures, the insights presented do not substantially advance current knowledge.

Conclusion

In summary, while the manuscript offers an informative descriptive analysis of potential biomarkers in endocrine-resistant breast cancer, these limitations in sample size, experimental work, lack of independent validation, and the redundancy of some findings preclude its publication in a high-impact journal like Nature Communications. Strengthening the study with a larger cohort, experimental validation of biomarkers, and a validation set would significantly enhance its impact and suitability for such a journal.

We acknowledge the previous work in the field that were consistent with some of our findings. We consider this a strength, validating the use of Ki67 biomarker approach used in the NeoPalAna trial. In addition to the analyses of pre-treatment samples, on-treatment biopsies analyses were uniquely presented with the goal to identify potential drivers associated with proliferation on ANA/PAL, which highlighted the existence of inflammatory and immune suppressive pathways including ADAR, IRAK1, and IDO1 (Fig. 4F). Our study also highlights the interplay between tumor cell intrinsic and immune microenvironment in association with resistance. In consideration of the comments raised by the other Reviewer(s), we have integrated our proteogenomic data for a more comprehensive view of the landscape underlying CDK4/6i-resistance (at baseline and at C1D15) and in response to these cell cycle inhibitors (looking at C1D15 vs baseline alterations). Notably as raised above, we validate targeting interferon/JAK-STAT across resistant *in vitro* models. Explorable therapeutic strategies in this resistant setting are summarized in New Fig. 4K. Lastly, we derive a resistance gene signature from our study, which demonstrates high sensitivity and specificity in stratifying poor versus good responders from baseline biopsies and of which is associate with patient prognosis validated across independent patient cohorts.

Reviewer #2 (Remarks to the Author):

Biomarkers of response to neoadjuvant palbociclib plus anastrozole in endocrine-resistant estrogen 1 receptor-positive/HER2-negative breast cancer

Kong et, al described a translational study based on the data collected from the NeoPalAna trial in which stage II/III ER+/HER2- BC patients were treated with ANA/PAL. Responding patients were compared with the resistant ones, on multiple -omics levels, with the purpose to understand the mechanism of resistance and to seek for response biomarkers. The study highlighted several dysregulated oncological pathways, different immune microenvironment, and protein IDO1 which is associated with PAM50 subtype, in patients who did not respond to ANA/PAL.

The paper is well presented, and the data is valuable for audience in oncology research community. The conclusion made so far has been properly underpinned by the analysis results. However, the authors have primarily focused on innate resistance mechanism while barely touched response biomarkers, as opposed to what its title suggested. The novelty of the conclusions is limited. The data could be further analysed to deliver its full potential.

We appreciate this Reviewer's acknowledge of our study and how it provides a valuable resource for the community. Please find our major additions to the study highlighted above.

Main points:

1. The study has a classical design: patients were categorised into responding and resistant cohorts, while samples were taken at pre-treatment and post-treatment, collectively forming a two-by-two data structure. Throughout comparisons have been made between the responding and resistant cohorts, at each individual time point, while little was compared between the pre- and post- treatment, within each patient cohort. The authors should characterise the changes at transcriptomics and proteomics level following ANA/PAL, and define the CHANGES that differentiate the responding vs. resistant patients.

In New Fig.5 (and below), we analyze treatment induced changes in gene expression in C1D15 versus Baseline samples across Sensitive and Resistant groups. New Fig. 5A highlights the PAM50 subtype changes of available paired samples. We then investigate the DEGs between C1D15 and BL samples in both Sensitive and Resistant tumors and identify unique and shared DEGs (adj- $p < 0.05$, log 2 fold change > 1 or $< -1.$, New Supplementary Table 7). The UpSet plot (New Fig. 5B) shows 3,032 genes that were uniquely upregulated in Resistant cases and 735 in Sensitive cases, with 1,208 shared. For downregulated genes, 909 were unique to Resistant cases and 613 to Sensitive cases, with 591 shared. We identified key cell cycle genes (including *CCNB2*, *AURKB*, *E2F1* and *E2F2*) that were suppressed at C1D15 in Sensitive, but not Resistant samples (New Fig. 5C).

We then performed GSEA on Sensitive C1D15 vs Baseline and on Resistant C1D15 vs Baseline samples, which revealed similar upregulated and downregulated pathways (New Fig. 5D). These findings suggest that there may not be one dominant pathway that is altered that may mediate response, but rather the degree of pathway activation at baseline and changes following treatment (and contribution by individual genes). To this, we investigated gene expression of pertinent oncogenic pathways in each group and at different timepoints. As also presented in New Fig. 3E (Baseline comparison), genes and pathways mapped to cell cycle and interferon were elevated (New Fig. 5E). The density plots of genes (normalized by mean Z-score) show that although both Sensitive and Resistant tumors undergo a comparable leftward shift in gene expression (ie. downregulation), however Resistant tumors retain relative pathway activation.

A**B****C****D****E**
2. The authors have rightly pointed out that using ki-67 as a surrogate for treatment response has its limitation. While acceptable in clinical trial design, this has a more prominent impact in biomarker discovery. In this study ki-67 acted as a perfect biomarker for response, leaving no chance a novel treatment-specific biomarker to be discovered. The authors should introduce objective response and other clinical endpoints for response biomarker discovery, followed with exploration on the relationship between the identified biomarkers and ki-67.

We thank the Reviewer for this point. Our initial attempts to find biomarkers related to Ki67 in our initial submission was highlighted in Old Fig. 4G (now New Fig. 4F). To summarize, Ki67_{C1D15} highly correlated with various markers involved in the cell cycle (*CCNE1*, *CCNA1*, *CCNA2*, *CDK6*, *PLK1*), and key translation factors (*EIF4EBP1*, *EIF4E2*, *E2F2*). In our trial design, patients who had persistent proliferation on ANA/PAL on C1D15 (Ki67_{C1D15} >10%) went off trial, therefore no clinical response data is available for these patients. While not a clinical response biomarker, C1D15 on treatment Ki67 is a measure of the anti-proliferative effect of ANA/PAL. Based on Ki67_{C1D15} level, we were able to categorize tumors that are sensitive vs resistant to ANA/PAL for the purpose of biomarker discovery.

In line with the other Reviewer(s) comments, in our revision we derived a baseline signature predicting response to palbociclib + endocrine therapy based on the C1D15 Ki67 response. We utilized both the 1) NeoPalAna ET-R cohort and 2) to increase our sample size, our previously published NeoPalAna initial cohort (NeoPalAna Initial), of which microarray gene expression data from pre-treatment tumors was available for 31 patients (26 Sensitive and 5 Resistant) (PMID 28270497).

We first assessed for similarity between the Resistant tumors in NeoPalAna ET-R and Initial cohorts. DEGs of interest from the NeoPalAna ET-R (*BRCA2*, *EZH2*, *IDO1*, *CCNE1*, *IRF4*, *CD48*, *CD38*; Fig. 3D), were also found to be significantly upregulated or near significantly upregulated in association with Resistance to ANA/PAL in NeoPalAna Initial (New Supplementary Fig. 4A, and below). Unbiased GSEA of Resistant vs Sensitive patients in the NeoPalAna Initial also identified upregulation of shared pathways including E2F targets, G2M Checkpoint, Interferon gamma response, Interferon alpha response, IL6-JAK-STAT, mTORC1 signaling, and MYC targets; relevant downregulated pathways included Estrogen response early (New Supplementary Fig. 4B, and below). These data indicate that two NeoPalAna cohorts share similar biomarkers of response to ANA/PAL.

Using the NeoPalANA ET-R and Initial cohort for discovery/training, we nominated 33 genes, among which incorporated key regulators of the cell cycle, oncogenic signaling, cellular proliferation, and mediators of immune

and interferon response as part of our NeoPalAna resistance signature (New Fig. 9A-D). We generated Receiving Operating Characteristic (ROC) curves and compared the performance of our gene signature to that of the baseline Ki67 by IHC. In NeoPalAna ET-R, the Area Under Curve (AUC) for Ki67 was 0.824 (95% CI 0.637,1) while the AUC for the NeoPalAna resistance signature was 0.967 (95% CI 0.902,1) (New Fig. 9E). Adding Ki67 to our gene signature did not improve the performance, with an AUC of 0.967 (95% CI 0.902,1). A greater AUC suggests a higher ability for test discrimination, with conventional standards 0.7-0.8 being considered acceptable, 0.8-0.9 as excellent, and >0.9 as outstanding (PMID 20736804). In NeoPalAna Initial, the AUC of Ki67 was 0.6 (95% CI 0.372, 0.828) while the AUC of the gene signature was 0.992 (95% CI 0.971,1), and combining the two 1.0 (95% CI 1, 1) (New Fig. 9F). Overall, the NeoPalAna resistant gene signature demonstrated high sensitivity/specificity in distinguishing between Sensitive and Resistant tumors.

For an external validation cohort, we collaborated with Dr. Erik Knudsen's group (Roswell Park) for an independent analysis of their "Real world" cohort (PMID 37704753), where targeted RNA-seq analysis of pre-treatment samples and clinical outcomes were available for 151 metastatic HR+/HER2- breast cancer patients were treated with first-line CDK4/6i (~90% palbociclib) in combination with hormonal therapy (115 with aromatase inhibitors +/- prior endocrine therapy, 36 Fulvestrant). We examined whether our resistant gene signature was prognostic for progression-free survival. Blinded with the clinical outcome data, the WashU group (TK, JL) applied the NeoPalAna resistance gene signature on pre-treatment RNA-sequencing data and assigned with low, medium, or high gene signature scores. An independent statistician (JW) from Dr. Knudsen's group then performed K-M analysis of patients stratified by the gene signature score. A higher resistant gene signature score was significantly associated with shorter PFS (HR = 2, $P = 0.012$) in the overall patient population (New Fig. 9G left panel, and below). Similar poorer outcome with shorter PFS, was seen in patients who received AI plus CDK4/6i (New Fig. 9G middle panel) and a trend in those that received CDK4/6i with fulvestrant (New Fig. 9G right panel), although fewer patients received this regimen. A higher gene signature was also associated with a worse overall survival (New Fig. 9H, HR = 2, $P = 0.03$). The analysis conducted in the NeoPalAna Initial and the independent "Real world" Metastatic cohort provided further validation of resistant biomarkers identified from the NeoPalAna cohorts.

3. Genomic, transcriptomic, proteomic, and cellular data were present separately without any attempt to provide an integrated view.

Minor points:

1. The authors have associated -omics changes with *ki-67* (267-290), but only provide description on a handful of selected gene/proteins. A better overview at -omics level, maybe in the format of correlation network topology and its changes, will help the audience understand the systematic omics-landscape shifting related to cell-proliferation as a result of ANA/PAL treatment.

We thank the reviewer for major point 3 and minor point 1. First, we acknowledge our limitations of sample collection and processing. Since patients had already undergone two breast biopsies (one at diagnosis, the other one at approximately 6 weeks on standard of care neoadjuvant endocrine therapy for Ki67 testing) before enrolling in the trial, we did not require research biopsies except at C1D15 when Ki67 is needed for treatment response evaluation. However, patients did consent to research use of archival tumor samples from biopsies occurred at diagnosis (Baseline) and for Ki67 testing on standard of care endocrine therapy (namely C1D1). Because of the limited tumor material from these biopsies, especially the archival tumor biopsies, serial samples were not available for all patients. Sufficient tumor samples for both genomic and proteomic analysis were only possible at C1D15. We reorganized the previous figures so the proteomic analysis were presented side by side of the RNA-seq analysis (New Fig. 4G, H, and below) and highlighted shared pathway enriched in Resistant vs Sensitive tumors at both the RNA and protein levels (New Fig. 4I). We observe a high degree of correlation (Spearman $r = 0.95$) between protein and RNA scores for these pathways.

For integrated proteogenomic analysis, we examined shared genes and proteins across top oncogenic pathways across the 14 samples that underwent both RNA-seq and mass spectrometry at C1D15. The highest RNA-protein correlations were observed in interferon and estrogen response pathway genes, whereas more discordance was observed in MYC, cell cycle, and DNA repair pathway genes (New Fig. 4J), which may suggest a role of additional post-transcriptional or post-translational processes at C1D15 in response to therapy.

To summarize the biomarker findings, we provided a schematic diagram (New Fig. 4K) which highlighted pertinent signaling pathways enriched in Resistant tumors and potential therapeutic strategies. We validated upregulated Interferon/JAK-STAT signaling in cell culture models of ER+ breast cancer cells resistant to palbociclib or abemaciclib, which were responsive to JAK2 inhibitors (New Fig. 6). Similarly, two PDX ER+/HER2- organoid PDX models that were resistant to palbociclib responded to the JAK2 inhibitor, pacritinib.

2. The number of DEGs are large. Maybe the limited sample size on the homogenous responding patients played a role? PCA showed heterogeneous omics profile in different BC subtypes. Is there a subtype-specific pattern in the DEG?

As this Reviewer points out in the PCAs (New Fig. 3C, 4C), three Resistant samples (Basal subtype) diverge from the other samples. In our original heatmap (Old Fig. 4F, now New Supplementary Fig. 5), the upregulated DEGs of interest are more elevated in these three samples. As shown in Supplemental Fig. 4, we do see conserved DEGs in Baseline samples in the NeoPalAna Initial cohort as NeoPalAna ET-R. However, given the limited sample size of Resistant samples in NeoPalAna ET-R, we have elected to include all samples to maximize power.

Despite heterogeneity, the derived 33-gene NeoPalAna resistance signature was able to capture shared features and demonstrated greater homogeneity across both NeoPalAna cohorts in PCA. (New Fig. 9C, D).

3. ANA/PAL-resistant and ANA/PAL-R have been used in a mixed way (234-235, and more).

We have now improved the consistency of nomenclature.

Reviewer #3 (Remarks to the Author):

In the manuscript reviewed, Kong, et al. utilizes the NeoPalAna endocrine-resistant cohort to explore the potential benefits of adding a CDK4/6 inhibitor to endocrine therapy (ET). Their findings demonstrate that combining palbociclib (PAL) with anastrozole (ANA) effectively suppresses cell proliferation resistant to ANA monotherapy, achieving complete cell cycle arrest (CCCA) in 57.6% of ET-resistant ER+/HER2- breast cancer (BC) cases by cycle 1, day 15 of treatment (C1D15). They then investigated potential mechanisms of resistance to the ANA/PAL combination, through a detailed genomics analysis of sensitive and resistant tumors. While no major genetic differences were identified, ANA/PAL-resistant tumors exhibited upregulation of cell cycle genes, suppression of G2M checkpoint genes, and activation of MYC and mTOR signaling pathways—consistent with continued proliferation (Ki-67 positivity). Most interesting is that the PAM50 classification was able to separate responders and non-responders. Additionally, some differences were observed in immune cell fraction induction (via CIBERSORT) such as presence of higher T-cell-inflamed TME and IDO1 levels in resistant tumors which could warrant the use of immune checkpoint inhibitors. Overall, despite the limited cohort size, this dataset represents a valuable, well-characterized resource for advancing research in ET resistance and potential therapeutic strategies in ER+/HER2- BC. Moreover, the results provide significant starting point for a biomarker panel that can identify responders to ANA/PAL and those who will need other treatments. Importantly, the genomic data suggests alternative paths for these resistant individuals.

We thank this Reviewer for their acknowledgeable of our dataset and work in better understanding treatment responses to CDK4/6 inhibitors.

Questions for authors:

1. The primary findings of significant clinical interest are: that 1) at baseline (BL), luminal A tumors are uniformly sensitive whereas HER2+/Basal were insensitive with Luminal B tumors being split in sensitivity and resistance; 2) higher BL Ki67 is clearly associated with resistance, and 3) evidence for immune activation in resistant tumors at baseline. However, simple clinical characteristics can also discern responders and non-responders. It would be helpful to address whether the molecular characteristics (e.g. taking the best biomarkers for response, like Ki67) would improve the prediction of response.

As discussed earlier in response to previous reviewers' comments, clinical characteristics were insufficient to discern responders vs non-responders (Table 1).

Based on the altered proteogenomic findings, we attempted to derive a baseline gene signature predicting response to palbociclib + endocrine therapy. We utilized both the 1) NeoPalAna ET-R cohort and 2) to increase our sample size, our previously published NeoPalAna initial cohort (NeoPalAna Initial), of which microarray gene expression data from pre-treatment tumors was available for 31 patients (26 Sensitive and 5 Resistant) (PMID 28270497).

We first assessed for similarity between the Resistant tumors in NeoPalAna ET-R and Initial cohorts. DEGs of interest from the NeoPalAna ET-R (*BRCA2, EZH2, IDO1, CCNE1, IRF4, CD48, CD38*; Fig. 3D), were also found to be significantly upregulated or near significantly upregulated in association with Resistance to ANA/PAL in NeoPalAna Initial (New Supplementary Fig. 4A, and below). Unbiased GSEA of Resistant vs Sensitive patients in the NeoPalAna Initial also identified upregulation of shared pathways including E2F targets, G2M Checkpoint, Interferon gamma response, Interferon alpha response, IL6-JAK-STAT, mTORC1 signaling, and MYC targets; relevant downregulated pathways included Estrogen response early (New Supplementary Fig. 4B, and below). These data indicate that two NeoPalAna cohorts share similar biomarkers of response to ANA/PAL.

A**B**
Using the NeoPalANA ET-R and Initial cohort for discovery/training, we nominated 33 genes, among which incorporated key regulators of the cell cycle, oncogenic signaling, cellular proliferation, and mediators of immune and interferon response as part of our NeoPalAna resistance signature (New Fig. 9A-D). We generated Receiving Operating Characteristic (ROC) curves and compared the performance of our gene signature to that of the baseline Ki67 by IHC. In NeoPalAna ET-R, the Area Under Curve (AUC) for Ki67 was 0.824 (95% CI 0.637,1) while the AUC for the NeoPalAna resistance signature was 0.967 (95% CI 0.902,1) (New Fig. 9E). Adding Ki67 to our gene signature did not improve the performance, with an AUC of 0.967 (95% CI 0.902,1). A greater AUC suggests a higher ability for test discrimination, with conventional standards 0.7-0.8 being considered acceptable, 0.8-0.9 as excellent, and >0.9 as outstanding (PMID 20736804). In NeoPalAna Initial, the AUC of Ki67 was 0.6 (95% CI 0.372, 0.828) while the AUC of the gene signature was 0.992 (95% CI 0.971,1), and combining the two 1.0 (95% CI 1, 1) (New Fig. 9F). Overall, the NeoPalAna resistant gene signature demonstrated high sensitivity/specificity in distinguishing between Sensitive and Resistant tumors.

For an external validation cohort, we collaborated with Dr. Erik Knudsen's group (Roswell Park) for an independent analysis of their "Real world" cohort (PMID 37704753), where targeted RNA-seq analysis of pre-treatment samples and clinical outcomes were available for 151 metastatic HR+/HER2- breast cancer patients were treated with first-line CDK4/6i (~90% palbociclib) in combination with hormonal therapy (115 with aromatase inhibitors +/- prior endocrine therapy, 36 Fulvestrant). We examined whether our resistant gene signature was prognostic for progression-free survival. Blinded with the clinical outcome data, the WashU group (TK, JL) applied the NeoPalAna resistance gene signature on pre-treatment RNA-sequencing data and assigned with low, medium, or high gene signature scores. An independent statistician (JW) from Dr. Knudsen's group then performed K-M analysis of patients stratified by the gene signature score. A higher resistant gene signature score was significantly associated with shorter PFS (HR = 2, $P = 0.012$) in the overall patient population (New Fig. 9G left panel, and below). Similar poorer outcome with shorter PFS, was seen in patients who received AI plus CDK4/6i (New Fig. 9G middle panel) and a trend in those that received CDK4/6i with fulvestrant (New Fig. 9G right panel), although fewer patients received this regimen. A higher gene signature was also associated with a worse overall survival (New Fig. 9H, HR = 2, $P = 0.03$). The analysis conducted in the NeoPalAna Initial and the independent "Real world" Metastatic cohort provided further validation of resistant biomarkers identified from the NeoPalAna cohorts.

2. There were issues that the authors do not comment on such as the relevance of T stage and more interestingly, race being significantly associated with response (Table 1). The observations are limited by the cohort size but how does this compare with other datasets? For example, do white patients respond better and Asian patients do not? Importantly, all patients underwent resection after the cycles of ANA/PAL and I assume that Table 2 represents these data, but surprisingly, there was no attempt to address whether any of the biomarkers were correlated with this important endpoint. This seems like a major flaw that is easily amenable in a revision. For example, there were PAM50 shifts at C1D15. Did any of those with shifts correlate with CR or PR?

We added the clinical variables including race and T stage found to be significantly associated with Ki67 response in Table 1 in the result section. However, these results should be interpreted with caution due to the small sample size, particularly race which only included 2 American Indian, 3 Black, and 3 Other, while the majority being white (N=25). Although all 3 Asian patients were ANA/PAL-resistant, 2 had LumB and 1 had HER2E BC, which disproportionally selected the high-risk disease by chance. Notably in the meta-analysis that included 4 first-line randomized trials of CDK4/6i/ET (n=2499), PFS benefit of CDK4/6i was observed in both Asian (n=492, HR 0.39 (95% CI 0.29–0.51, P<0.0001) and non-Asians populations (n=2007, HR 0.62 (95% CI 0.54–0.71, P<0.0001) (PMID 30465154). In MonarchE, DFS benefit was observed in both Asian and White populations (PMID 36493792). In the adjuvant setting, both stage II or III tumors benefited from adjuvant CDK4/6i in the Menarche E and in the NATALEE trial (PMID 39442617).

In the NeoPalAna trial, patients who had Ki67 >10% went off the trial, either proceeded to surgery or underwent neoadjuvant chemotherapy. Therefore, only patients who had Ki67 response completed 4-5 cycles of ANA/PAL for clinical response assessment. As a result, we could not address PAM50 switching in relation to clinical response since all patients had LumA except 1 at C1D15.

3. While these BL observations are noteworthy and potentially clinical important, the impact of the C1D15 seems less clear but has significant biological interest. That there were such significant expression changes only 15 days after initiation of ANA/PAL treatment is intriguing and is not likely to be related to clone fluxes rather to direct transcriptional effects of the therapy. Yet, other than a descriptive narrative, the authors seemed to have avoided much conjecture as to the potential biology, nor did they integrate these data into the final narrative. Moreover, Figure 4, which depicts the C1D15 data is poorly annotated so it is not clear to this reviewer how to interpret some of the data. For example, the labels for the X and Y axis in panel G does not explain what sensitive r means and what is the interpretation of why CCND1 and AKT1 is below the dotted line, This all seems like a wasted opportunity. The shift in PAM. That the immune markers for IFN signalling (Figure 4E) actually increased in C1D15 is of interest, perhaps the authors could, in the discussion, explore this a bit further. This seems to be contrary to the ciphersort results in Figure 6 where the C1D15 levels of the different immune subtypes do not appear to be going up. Again, this may be the confusion of what each of the figures are referring to – BL, or C1D15, For the patient tumors that were classified as ANA/PAL sensitive at C1D15, it appears that most of them maintained their relative sensitivity based on clinical response reported in Table 2. What were the three tumors that had stable disease? Did they have any specific characteristics (PAM50, grade, genetic or transcriptional signals etc) compared to the tumors that had partial or complete response?

In response to reviewers' comments, we added the analysis for treatment induced changes in gene expression profiles at C1D15 compared to pre-treatment timepoint, in Sensitive vs Resistant tumors. The clinical response data shown in Table 2 included patients who had Ki67 response at C1D15 and completed 4-5 cycles of ANA/PAL before surgery. The 3 tumors had SD is not particularly different from the group, all being non-Hispanic White, with invasive ductal carcinoma, 2 grade 2 and 1 grade 3 tumor. One patient had LumA, the other two patients did not have sufficient diagnostic tumor material for PAM50 subtyping. As discussed earlier, per study design, we do not have clinical response data for those with Ki67 over 10% at C1D15, therefore unable to do the analysis by response. We have now updated our results and discussion to further expand upon these findings and our newly added data including JAK/STAT targeting, C1D15-BL comparisons, and our derived baseline resistance signature.

4. Similarly, since PAM50 subtype switching is observed in the tumors before and after treatment, it would be simpler to visualize the data in a manner (plot or table) that shows the subtypes before and after treatment for each tumor instead of just a Venn diagram showing the subtypes pre and post treatment in Fig. 3A and 4A. Is there any association with PAM50 subtyping and response at C1D15? Also Table 1 has “PAM50” notes as “PAM5”.

Thank you for this suggestion. We have provided a new Sankey diagram in New Fig. 5A that shows the subtype changes at C1D15 compared to Baseline for the tumors that had serial biopsies collected. New Fig. 5 now also highlights transcriptional changes between C1D15 and Baseline. As highlighted in Fig. 4A, 18/19 Sensitive tumors at C1D15 were LumA (1/19 normal). In contrast, all non-LumA subtype (HER2E, Basal) samples were Resistant, with findings being consistent with known subtype associations with CDK4/6i resistance. Resistant and Sensitive C1D15 tumor subtypes were statistically distinct ($P=0.0014$) by 2-tailed Fisher's exact test. Thank you for the “PAM5” annotation; we have now corrected this.

5. Since WES data was generated, have the authors looked for mutation calls across of genes vs just the 83 genes reported here (Fig.2)? Were any other highly occurring mutations observed in the cohort? Were any of them associated with resistance or sensitivity? What are the criteria for selection of genes shown in copy number heatmap in Fig.2B? All the genomic analysis does not appear to be comprehensive enough to conclude that there is no mutation of copy number changes associated with response.

Per this Reviewer's comment, we looked at the mutation burden with less stringent calling across all the genes and plotted the top 50 frequently altered genes (Reviewer figure below). We observed that *LRP1B* was altered in 6/18 Sensitive tumors and 0/8 Resistant tumors. Other notable genes were *HLA-A* (1S, 4R), *ARID1A* (4S, 1R), and *MAP2K4* (4S, 0R).

Due to the small sample size, the CNV analysis at our initial submission focused on targeted panel of cell cycle genes and those that annotate canonical oncogenic drivers previously implicated in CDK4/6i resistance, and as such we have left this initial figure unchanged in our revision. We however, do agree with the reviewer that a more unbiased presentation could be beneficial and performed differential analysis with the Limma statistical package to identify genes with altered CNVs. This is now presented in New Supplementary Fig. 3 and in New Supplementary Table 3. We then took the top 500 genes and performed enrichment analysis, which revealed alterations in pathways including DNA repair, PI3K/AKT/mTOR signaling and E2F targets (New Fig. 2C) and have updated the text accordingly. However, these results should be interpreted with caution due to the small sample size.

6. *The limited size of the dataset was probably why no significance could be established on the relevance of the PIK3CA mutation incidence in ANA/PAL-resistant vs sensitive cases. Are there any other publicly available datasets that could be analyzed for this? The authors referred to other similar studies with biomarkers but there did not seem to be a systematic comparison of the biomarkers across these studies.*

We included the data from the NeoPalAna Initial, in which all *PIK3CA* mutated primary ER+ breast cancers were responsive to the antiproliferative effect of ANA/PAL, while 6 of the 28 *PIK3CA* WT tumors were resistant (PMID 28270497). These findings are not particularly surprising as *PIK3CA* mutation is often associated with LumA subtype in the primary breast cancer. This however may differ in the context of metastatic disease, in which *PIK3CA* mutation is often a poorer prognostic marker in HR+ breast cancer (PMID 32067679).

7. *In Fig.2A, the authors should describe what the S and R barplots on the right side are. Is it the percentage of all samples with the mutation in each category? The barplot in the top panel of the figure also has no y axis label. In Fig.2B please add in legend for sensitive/resistance sample annotation at the side of the heatmap.*

We apologize for the confusion. The left bar plot represents frequency of mutations seen in this cohort (27 samples). The right S/R bar plots represent frequency of alterations in S (orange) and R (pink) samples within all S (n = 18) and R (n = 8) samples. These plots were shown for visual comparison. We have clarified the bar plots description in the figure legend. We have added the legend for Fig. 2B.

8. *“Top positively enriched DEGs..” described in line 235 don’t really all appear to be “top” DEGs in the volcano plot in Fig.3D by fold change or *fdr*. Please describe the “top” DEGs and rephrase the rationale for highlighting the genes listed.*

We have updated our phrasing to: “Statistically significant, positively enriched DEGs of interest in Resistant BCs included tumor-associated effectors *CCNE1*, *IDO1*, *IRF4*, *BRCA2*, *EZH2*, and immune cell markers such as *CD38* and *CD48* (Fig. 3D) among others (Supplementary Table 4).” We followed up with additional analysis on some of these identified effects/pathways at Baseline (eg. cell cycle mediators, *IDO1*) throughout the manuscript.

9. *In Figure 3E and 4E, please include *fdr* or *p* value for the pathway enrichment analysis. Only NES is shown. The text (lines 236-240) reports the pathway enrichment analysis but don’t really highlight the relevance of this analysis. Please rephrase text accordingly.*

We have now provided the *P-values* for Fig. 3E, 4E, New Fig. 4H, and New Fig. 5B in a new Supplementary Data file. We have expanded on these findings in the results and discussion sections to highlight these pre-existing proliferative hallmarks prior to the start of ANA/PAL and their association with poor treatment outcomes.

10. *In PCA plots in Fig.3C and 4C, the basal subtype tumors always separate out from all other tumors (irrespective of being sensitive or resistant) on PC1. Would you see a separation of the resistant and sensitive tumors if you remove the three basal tumors from the analysis?*

The PCA plots were performed on top DEGs derived from RNA-seq analysis. We utilized the same DEGs, removed the three basal samples and reran PCA (see figure below). In the Baseline samples, we observed that the clustering of the Sensitive samples largely remained, suggesting relative biological homogeneity of these Sensitive samples, whereas Resistant samples demonstrated greater biological heterogeneity. At C1D15, there was reduced separation between the Sensitive and Resistant tumors following removal of the basal samples. These findings reiterate that in the NeoPalAna ET-R cohort, the PAM50 subtype is a crucial factor in predicting response to CDK4/6 inhibitors, which is supported in Table 1, and is consistent with the current literature highlighted by Reviewer 1.

11. In Fig.5D and 7A please mention what the change in the circle size represent.

In the Old Fig. 5D proteomics bubble plot, the relative size of the circle represented the number of genes found in each gene signature. In our revision, we have reformatted Fig. 5D (now New Fig. 5H) into a bar graph for consistency with the Fig. 5E mRNA bar graph.

The circles in old Fig. 7A (now new Fig. 8A) represent the spearman coefficients between two genes, with a larger relative size corresponding to a larger spearman coefficient. We have updated this in the figure legend.

12. In fig.6B, despite seeing a higher level of CD8+ T cells and M1 macrophages at baseline, the tumors still acquire resistance to ANA-PAL treatment. Moreover, at C1D15 there is no significant difference in the levels of T cells, NK cells, M1 and M2 macrophages between sensitive and resistant tumors. The interpretation of these results can be commented on more in the discussion. For example, looking at absolute levels of immune cells post treatment may not be indicative of response in these tumors?

Our data suggests that the highly proliferative endocrine resistant HR+ breast cancer cells are enriched with the expression of inflammatory and interferon pathway genes. The findings of enrichment of immune checkpoints and immune inflamed but suppressive microenvironment of CDK4/6i-resistant breast cancers were consistent with previous reports. Interestingly, our data comparing treatment induced changes in immune cell populations between Sensitive and Resistant tumors suggest CDK4/6i may modulate the immune microenvironment differently. Although our data could not provide an underlying etiology that explains such interaction, the association of CDK4/6i-resistant tumors with increased expression of immune checkpoint proteins and an inflamed but immune suppressive microenvironment could be exploited for opportunities to use immune checkpoint inhibitors in future clinical trials. This hypothesis is supported by recent reports that a subset of high grade HR+/HER2- breast cancer with increased PD-L1 expression and tumor infiltrating lymphocytes could benefit from the addition of immune checkpoint inhibitors (KEYNOTE 756 and CheckMate 7FL). We have updated our discussion accordingly.

13. The first p-value line in Suppl. Fig S2A, appear to be marked wrong. Are the authors comparing NeoPalAna ET-resistant cohort (pink) compared to that of the Z1031_POL ET-sensitive cohort (yellow)?

The TP53 mutation frequency was comparable in the Sensitive and Resistant groups in the NeoPalAna ET-R. Our intention in Supplementary Fig. 2A was to compare the overall NeoPalAna ET-R cohort to the Z1031 ET-Sensitive group to highlight the higher frequency of TP53 mutation in the endocrine resistant population accrued in NeoPalAna ET-R.

Reviewer #4 (Remarks to the Author):

We thank the Reviewer for their time in reviewing our manuscript and for the helpful comments.

Response to Reviewers' Comments

We thank the reviewers for the insightful comments. Please see our point-by-point response below.

Reviewer #1 (Remarks to the Author)

The new data and experiments improve the overall quality of the manuscript, addressing the majority of the concerns raised by this reviewer. Although the number of patients can't be changed, the validations and the experiments make the whole work more convincing and suitable for this journal.

Response: We thank the reviewer for the positive assessment of our revised manuscript.

Reviewer #2 (Remarks to the Author)

Substantial changes have been made to the manuscript following the reviewers' comments. With additional datasets and new analysis, the authors were able to characterise -omics landscape related to cell proliferation and changes following CDK4/6i. The comparison between resistant and response samples, at molecular levels and functional levels, provide insight to the mechanism underpinning treatment response. The new 33 gene baseline model is a good add-on.

The sample size unfortunately limited the depth of this research, nonetheless I am satisfied with the level of evidence underpinning the main conclusion of the paper, and found that the major concerns from the reviewers have been addressed to a satisfactory level.

The only addition I would like to suggest is to include in this manuscript is a summary table of RNA/protein data related to the 33 gene (BL predictors) in the supplementary material.

Response: We thank the reviewer for the positive feedback and guidance to further strengthen our manuscript. As requested, we have added a summary table of the 33 genes, including the area under curve (AUC) for each candidate in predicting resistance to anastrozole in combination with palbociclib in the NeoPalAna ET-R and NeoPalAna Initial cohorts, along with their annotated biological functions (new Supplementary Table 8). In addition, Fig. 9B has also been updated to highlight the canonical, biological function of each candidate.

Reviewer #3 (Remarks to the Author)

On the whole, this revision of the work is improved. Integration of RNA-seq and mass spectrometry analysis and Transcriptomic changes between C1D15 and baseline in Resistant versus Sensitive tumors improve the paper significantly. Moreover, the addition of the validation cohort was very helpful.

Response: We thank the reviewer for the positive assessment of our revised manuscript.

However, there are several issues that need some attention:

The additional analysis with the 33 genes signature on the three datasets indeed looks promising and makes good utilization of all the data generated. The authors however, should clearly elaborate on how the 33 genes were derived/nominated based on the NeoPalAna ET-R and NeoPalAna Initial data. The number 33 seems arbitrary and without any rationale. Were they picked because they were overlapping genes in both the dataset sensitive vs resistant comparisons or had molecular pathway relevance etc? In Fig.9B could you highlight the genes based on the different categories described (cell cycle, oncogenic signaling, proliferation, and immune and interferon response mediators etc). This is a major oversight.

Response: We thank the reviewer for the opportunity to clarify our methods for deriving the gene signature. We identified the signature based on both statistical evidence and *a priori* knowledge following a workflow described below. From the two training cohorts (the NeoPalAna ET-R and NeoPalAna Initial cohorts), candidate genes

were first unbiasedly selected based on fulfilment at least one of the following criteria: 1) Differentially expressed genes (DEGs) between Resistant and Sensitive groups from the NeoPalAna ET-R cohort, of which included overlapping DEGs from the NeoPalAna Initial cohort (915 DEGs from NeoPalAna ET-R identified using DEseq2 with adjusted p -value < 0.05 , of which 332 were shared with NeoPalAna Initial cohort identified using Limma with a p -value < 0.05 given the small sample size of Resistant samples); 2) annotation in key enriched molecular pathways identified through GSEA with FDR p -value < 0.05 including E2F targets, G2M checkpoint, Interferon alpha response, Interferon gamma response, MTORC1 signaling, MYC targets V1, and MYC targets V2, in addition to Hallmark Estrogen response early, Hallmark Estrogen response late, and KEGG cell cycle; 3) genes of interest including *MTOR* (not annotated in the Hallmark MTORC1 signaling gene set), key breast cancer markers (*ERBB2* and *ESR1*), and those mediating immunological pathways (Tumor inflammation signature and HLA genes). Logistic regression models were performed for ROC curve analysis for each candidate gene. The well-known biological and clinically validated markers, such as *ESR1* (estrogen receptor) and *ERBB2* (HER2), were included in the final signature despite having a weak AUC in the NeoPalAna Initial cohort (AUCs 0.55 and 0.59, respectively). These criteria resulted in generation of the 33-gene signature. 23 of 33 genes were significant in the differential expression analysis from the NeoPalAna ET-R cohort, highlighting that this dataset unbiasedly contributed substantially to the selection of genes.

Of the 33 genes in the signature, 27 were available in the metastatic validation cohort, which was profiled using the HTG EdgeSeq Oncology Biomarker Panel (covering 2,560 genes). Although six candidates (*ARPC5L*, *BCL11A*, *CDCA8*, *SKA1*, *TCEAL4*, and *TES*) were missing in this panel, the remaining genes in the gene signature retained sufficient discriminatory power to stratify patients and demonstrated meaningful associations with survival outcomes. We have updated the Methods section accordingly. A schematic of the workflow is provided below and in new Supplementary Fig. 6.

As requested, we have now provided a summary table of the 33 genes, including the area under curve (AUC) for each candidate in the NeoPalAna ET-R and NeoPalAna Initial cohorts, along with their annotated biological functions and workflow schematic (New Supplementary Table 8, New Supplementary Fig. 6, and below).

The discussion around figure 6 is rather confusing. While the expression analysis and western blot data on the MCF7 *in vitro* analysis looks promising the *in vitro* drug response data is not convincing. For example - Authors justify the use of JAK/STAT inhibitors based on the expression analysis on Palbo-R vs parental and Abema-R MCF7 vs parental cells. However there is no real difference (increased sensitivity) observed between parental and Palbo-R or Abema-R cell lines for any drug or dose (Fig.6H). In fact, it seems that the resistant cell lines are perhaps less sensitive than the sensitive line. The data just shows that all the 3 cell lines have increased cell death for increasing concentrations of most drugs - which would be expected? If the data does not show that Palbo-R or Abema-R MCF7 cells have increased sensitivity to JAK inhibitors compared to parental cells what are the authors trying to state with the data?

"Screening with these inhibitors and the JAK1 selective inhibitor itacitinib in parental MCF7 and CDK4/6i-resistant cells suggested a higher dependency on JAK2 than JAK1, with similar sensitivity across the cell lines (Fig. 6H)." That Pacritinib is more effective on higher doses tested (in all three cell lines) compared to other drugs could just be a function of concentrations tested in the experiment. That no cell death is observed in response to Itacitinib could just be inadequate concentrations tested in this experiment. In order to draw this conclusion of increased dependency on JAK2 vs JAK1 authors will have to normalize to the JAK molecule inhibitory effect of the drugs for a given concentration or perform gene knock out experiments to confirm the JAK1 vs JAK2 (and IRAK1) sensitivity.

For the *in vitro* drug data on organoids derived from PDX tumors (Fig.6K and 6L), the authors state that "Both organoid models showed reduced sensitivity to palbociclib (Fig. 9K) but were responsive to pacritinib at low micromolar doses (Fig. 9L), which was comparable to the response observed in MCF7 Palbo-R/Abema-R cells. These data support the potential importance of JAK/STAT signaling in CDK4/6i resistant BCs." Not sure if the data shown is enough to compare sensitivities between drugs and make this statement - these are two different drugs with two completely different mechanisms of action. So if WHIM43 for example has an IC₅₀ of 1.9µM to Pacritinib (JAK2 inhibitor) and 3µM to Palbociclib (CDK4/6 infinite) does it have any significant biological implication? Also figures 6K and 6L have been referred to wrongly in the text.

Response: Thank you for these insightful comments. As noted, there were no substantial differences in the sensitivity of the panel to JAK2 inhibitors between the parental and Palbo-R/Abema-R MCF7 cell lines. These data suggest that JAK-STAT pathway may not be the sole driver of drug resistance, but rather may also represent a downstream consequence of other signaling pathways. particularly in the *in vivo* breast cancer context where interplay with the tumor microenvironment exists. Further studies are warranted to clarify the role of JAK-STAT pathway upregulation in CDK4/6i-resistant breast cancer, particularly in the *in vivo* breast cancer setting where interactions with the tumor microenvironment can profoundly influence signaling dynamics and therapeutic response. For instance, suppression of the JAK-STAT pathway could attenuate both oncogenic signaling and immune modulators such as PD-L1 based on previous studies (PMID: 26155422). We have updated our discussion to reflect these points.

Thank you for the comments regarding JAK2/JAK1. We have revised our interpretation of the JAK2/JAK1 comparison.

With respect to defining sensitivity vs resistance to palbociclib or pacritinib in the PDX-derived organoid studies shown in Fig. 6K and Fig. 6L, it should be noted that *in vitro* IC₅₀ values cannot be directly compared across these two agents. Instead, they should be interpreted in the context of steady-state plasma concentrations achieved in patients receiving approved doses (~0.13-0.28 µM for palbociclib [PMID: 35890213]; ~12–13 µM for pacritinib [PMID: 33232476]). On this basis, the observed IC₅₀ of 1-2 µM for pacritinib falls well within the clinically relevant exposure range, whereas the ~3 µM IC₅₀ for palbociclib is far above concentrations achieved in patients and thus not therapeutically attainable. These data support the conclusion of sensitivity of the WHIM PDX organoid models being resistant to palbociclib but sensitive to pacritinib. Also as shown in Fig. 6I that pacritinib at 1 µM suppressed pSTAT3 signaling in MCF7 Palbo-R, consistent with our prior work in myeloproliferative neoplasm models where 1 µM pacritinib effectively inhibited STAT1, STAT3, and STAT5 phosphorylation in CD34⁺ progenitor cells (PMID: 37203407), demonstrating on-target JAK-STAT pathway suppression. Nonetheless, we agree that further mechanistic studies are warranted to delineate the precise pathways through which pacritinib exerts its anti-tumor effects in ER⁺/HER2⁻ CDK4/6i-resistant breast cancer.

We have updated the references to Figs. 6K and 6L in the text.

"Zero Interaction Potency (ZIP) drug synergy calculations of CDK4/6i and pacritinib at indicated doses." What are the doses? There is no description of how this synergistic experiment was performed and how should one interpret the colors etc. The legend was not helpful.

Response: We apologize for the lack of clarity in this figure. These synergy experiments were performed utilizing multiple drug doses by 2-fold serial dilution (palbociclib/abemaciclib doses: 0, 0.5, 1, 2, 4, and 8 μ M; pacritinib doses: 0, 0.16, 0.31, 0.63, 1.25 μ M). We have now included the doses in the updated figures, updated the scale bar, and updated the figure legend to include that green coloring represents a negative ZIP score (antagonism when ZIP < -10) and red coloring represents a positive ZIP score (synergism when ZIP > 10).

Authors should describe in results or legend data in figured better - for example in fig 6G -western blot - There is no mention of AKT and RB results.

Response: We have now expanded on these findings in the results section to highlight that 1) RB expression was retained, which is important since its loss is a well-established mechanism of CDK4/6i resistance in preclinical models and 2) elevated AKT phosphorylation to support augmented oncogenic signaling in resistant lines.

Reviewer #4 (Remarks to the Author)

Response: We thank this reviewer for their time in reviewing our manuscript.

Reviewer #5 (Remarks to the Author):

As a statistical reviewer, I will focus this review on the methodology of the clinical trial and the development of the molecular signature.

To summarize this work, the authors studied the mechanisms of resistance in ER+ breast cancer to the combination of anastrozole + palbociclib. Translational experiments are performed on samples of patients included in a single-arm phase II Simon's two-stage study. In this study, the combination is used in a neoadjuvant setting, for patients with large tumors (T2 or more) whatever the lymph node status and without metastatic disease.

The clinical study was stopped before the end of recruitment after the decision of the data and safety monitoring board for efficacy, which is an unusual situation, as Simon two-stage design allow to stop the study for futility, not for efficacy.

The primary endpoint is the rate of complete cell cycle arrest (CCCA) after 1 month of combination of anastrozole + palbociclib, assessed by Ki67% immunohistochemistry and centrally reviewed. The sample size is reproducible, though the hypotheses are very pessimistic (unacceptable rate of 5%, acceptable rate of 20% of CCCA).

In the analysis of the primary endpoint, there were 22/33 (66%) of patients that exhibited CCCA at C1D15.

Major comments:

- My main concern on that work is about the 33 genes signature that would be predictive of resistance to the combination of anastrozole + palbociclib (ANA+PAL). There is first a strong issue of reporting. The authors can look at the TRIPOD recommendations, that have been recently updated for AI-based prediction models (PMID:

38626948). Even though these recommendations are for dedicated work on prediction models (not integrated to a translational work), it could serve as a basis for the reporting of this part of the work. There is no info on the selection process of the variables (and their possible transformation) into the model, nor the nature of the model used to estimate the probability of being resistant (logistic regression model? Others models?).

- Then, probably the biggest issue, there is only 33 patients to develop this model (with 11 patients that presented the endpoint “resistance to ANA+PAL”) which is very problematic if genes are introduced as unique variables in the analysis (33 potential variables). For example, a very loose rule of thumb is 10 events for 1 variable in a logistic regression model (therefore in the case of 11 events, only 1 variable can explain the outcome). And finally, there is no correction of optimism when estimating the performances of the gene signature (e.g. AUC) by any technique (for example, bootstrap, see PMID: 38191193). The optimism is linked to the fact that the same data are used to choose the model and evaluate its performances.

Response: We thank the reviewer for highlighting the TRIPOD recommendations. We agree that the added detail and greater transparency of our study protocol and analysis would be beneficial for the community. As such, we have amended our study reporting of relevant topics according to the TRIPOD+AI checklist (New Supplementary Data 2).

We are aware of the rule of thumb that 10 observations are roughly needed to support one variable in traditional statistical modeling, and that high dimensional omics data suffers from the curse of dimensionality of “small N, big P” and calls for caution and handling of the high dimensionality. We selected each biomarker first through univariate analyses based on differential gene expression (DE) analysis (where we corrected for multiplicity in picking differential genes) and ROC analysis using the metric of AUC to assess the discriminative ability of each individual gene. We also complemented the selection process with *a priori* knowledge in the field. See details of candidate gene selection in our reply to Reviewer 3 and in the updated methods section. We then exploratively assessed the overall discriminative ability of the selected genes collectively in an unsupervised manner using sparse principal component analysis (Fig. 3C). To the end, we constructed the averaged Z-score of the selected genes (where all the genes were oriented the same direction to have higher expression) as the composite signature score to circumvent the issue of “curse of dimensionality”. In this revision, as suggested by the reviewer, we conducted additional bootstrapping analyses, as part of the internal validation, to report the optimism-corrected AUC as a robust evaluation of the performance of individual biomarkers we selected and the signature. We found the bootstrapping based results agreed greatly with the results we originally reported (New Supplementary Table 9, new Supplementary Data 3, and below).

We agree with the reviewer that the small sample size of the study may limit our ability to develop a robust and generalizable gene signature and have acknowledged this in our discussion. While we are fully aware of this limitation, unfortunately, small sample size is the reality for many studies, as it is for our study. We thus adopted the standard strategy universally adopted by most biomarker signature studies: we derived the signature using a training set, followed by a completely independent external validation. In the external validation experiment,

we were fully blinded to the patient phenotypes and clinical efficacy data. Using the RNA sequencing data, we derived individual patient signatures and provided them to the validation team, who independently conducted the association analyses to evaluate the predictive ability of the signature in their validation cohort. The rigorous external validation (Fig. 9G-H) supports the predictive value of the signature, though future large studies are needed to confirm and generalize these findings. Importantly, while the signature was derived using an intermediate endpoint (C1D15 Ki67, 10% cutoff), the external validation employed long-term outcomes (PFS and OS), making it a more rigorous test.

- Regarding the external validation of the score, there is several issues. First, the score has been developed to predict the ANA+PAL resistance (binary, yes or no) in the neoadjuvant setting, and is validated on another outcome (progression-free survival). This is therefore not a good external validation (for more information, see PMID: 38224968). Second, there is no information on how are chosen the thresholds “low”, “medium” or “high” for expression of the signature (quartile? distributions?). On the Kaplan-Meier curves, the 95% confidence intervals are not displayed, but when looking at the number of patients at risk, we can see that the subgroups are quite small.

Response: We appreciate the reviewer’s perspective. However, we believe that validating the signature against survival outcomes strengthens the analysis, as survival endpoints are universally regarded as the gold standard. While Ki67 provides an early readout correlated with long-term outcomes, the ultimate goal of our signature is to predict survival, making this validation both appropriate and clinically meaningful. Obtaining an external validation cohort in CDK4/6i-resistant populations is challenging, making the metastatic breast cancer validation dataset particularly valuable. The consistent identification of candidate mediators and predictors across cohorts reinforces the robustness and clinical relevance of our findings.

For the validation cohort, patients were stratified into tertiles according to their gene expression score (also updated in our methods section). We have also regenerated the Kaplan-Meier curves with the 95% confidence intervals for this reviewer (see below). Although the subgroups are relatively small as indicated by this reviewer, the survival data remain both statistically significant and clinically meaningful, particularly for this metastatic dataset.

- Regarding clinical response, only 21/34 patients that completed the 5 cycles had a reported clinical response. This means that this response rate is evaluated only in sensitive patients and not in resistant patients that stopped the treatment. This should be more transparent in the narrative. In addition, in an intention-to-treat analysis, which is recommended to avoid to inflate the effect of the treatment, the denominator of the response rate should be 34 and not only 21 as current.

Response: The Result section describing clinical response has been updated to clarify the response rate both in patients who completed all cycles of therapy and the ITT population.

- It is written that it was a RECIST evaluation (radiological evaluation based on target lesions), but the rhythm

of assessment is not defined. It is a response at a given time point (e.g. at surgery) ? Is it the best overall response on treatment? This point deserves some clarification.

Response: The definition of clinical response and the corresponding assessment timepoint have been updated in the Methods section.

Minor comments:

- The authors stated in the responses to the rebuttal letter that “As discussed earlier in response to previous reviewers’ comments, clinical characteristics were insufficient to discern responders vs non-responders (Table 1)”, which is not completely true, as tumor size (T3 versus T2) and grade (III versus I-II) were associated with resistance, and may have been included in the model that predicts resistance to ANA+PAL

Response: We apologize for this erroneous inclusion in the initial rebuttal letter. We would like to clarify that clinical characteristics including tumor size and stage were NOT included in the resistance gene signature.

- The CCCA seems to be used more and more in the neoadjuvant setting for breast cancer as an endpoint, but its relevance for patients could be debated. Notably, the rate of patients that managed to undergo surgery is probably a more meaningful clinical endpoint, even though it is understandable in the setting of this translational study that this endpoint has been chosen

Response: We agree with the reviewer that endpoints should reflect outcomes meaningful to patients. We selected complete cell-cycle arrest (CCCA; Ki67 \leq 2.7%) at Cycle 1 Day 15 (C1D15) as the primary endpoint because it is an on-target, early pharmacodynamic marker for CDK4/6 inhibition.

- The one-sided error rate is 10% for the computation of the sample size, therefore if the authors want to compute confidence intervals that mirror the sample size, it is not a 90% coverage but an 80% coverage that should be chosen. Anyways, as this is not a sample size based on confidence interval, it is not mandatory to reduce the coverage of confidence intervals

Response: While the confidence coverage level is not directly related to the signature itself, the reviewer is correct that the confidence interval accompanies the 1-sided error rate of 10% is usually $100 \times (1 - 2 \times \alpha)$ % and thus 80% coverage. We chose to report the traditional two-sided 90% confidence interval following the recommendation by Steiger (Psychological Methods 2004, Vol. 9, No. 2, 164–182, <https://doi.org/10.1037/1082-989X.9.2.164>, see the top texts on the Page 174 under Section “The Relationship Between Confidence Intervals and Hypothesis Tests—Choosing the Appropriate Interval”).

To conclude, from the statistical perspective, the paper does not meet the standards of a well-conducted clinical research study. I would expect that Nature Communications would require a better methodology for clinical research, as they do for basic science.

Response: We appreciate the reviewer’s thoughtful comments and have revised the manuscript to strengthen the scientific rigor. While certain limitations (e.g., sample size) are inherent, we have a transparent signature development process where candidate genes were selected based on statistical evidence and *a priori* field knowledge and most importantly, the findings were validated in an independent cohort in a blinded fashion. We believe these revisions enhance the transparency and methodological robustness of the manuscript and align it with the standards expected by *Nature Communications* for clinical and translational research contributions.

Reviewer #5 (Remarks to the Author):

Statistical review.

Regarding my main concern—the development of the signature—the authors have significantly improved the transparency of the reporting of their results. However, several issues remain:

Could you please annotate the code regarding correction for optimism of AUC? I am not sure to understand what have been done.

See 10.1136/bmj-2023-074819 “Bootstrapping is a resampling technique, where a bootstrap sample is created by randomly sampling (with replacement) from the original data. In the enhanced bootstrap, a model is developed (repeating the model building steps used to develop the model on all the data) in each bootstrap sample and its performance evaluated in this sample as well as the original dataset to get an estimate of optimism of model performance. This process is repeated many times and the average optimism calculated, which is then subtracted from the apparent performance.”

We have now annotated the R function implementing the over-optimism correction of AUC.

We performed the overoptimism correction via bootstrapping, with the following detailed steps:

- (1) Calculate the AUC of an individual gene (or a composite) using the original data, denoted by AUC_{original}**
- (2) Generate bootstrap data sets ($k=1, 2, \dots, K$ and K set at 1000) by random sampling with replacement.**
- (3) Within each bootstrap dataset k , we recalculate the AUC of an individual gene (or composite) as AUC_k**
- (4) Calculate the difference between a bootstrap AUC and the original AUC as $\Delta AUC_k = AUC_k - AUC_{\text{original}}$**
- (5) Calculate the averaged AUC difference across all the K bootstrapping datasets as ΔAUC . ΔAUC should presumably be negative, as we expect lower AUC from bootstrapped data as compared to using the original data.**
- (6) Calculate the corrected AUC by subtracting the optimism part from the original AUC: $AUC_{\text{optimism-corrected}} = AUC_{\text{original}} + \Delta AUC$**

The steps of optimism correction are now incorporated into the Supplemental Information.

Why does the methods section not mention that the signature is defined as the averaged Z-score of the genes, as stated by the authors in their rebuttal letter?

The averaged Z-score is used as the composite signature. However, we are in the process of a patent application and thus prefer not to disclose this detail of constructing the signature.

Why was bootstrap validation performed on each gene independently, rather than on the composite model using the signature and/or Ki67 as the dependent covariate?

The performance of the composite relies on the performance of the individual genes. Thus, we started by evaluating the individual genes. We now performed a similar bootstrapping validation on the composite signature without and with ki67 and reported the optimism-corrected AUCs for ki67 alone, for composite signature alone, and for ki67+composite signature in the Results section.

The manuscript would also benefit from clarifying some points:

Line 213, page 9: The phrase “in an unbiased manner” should be removed. The authors incorporated expert knowledge in the selection of covariates, which, while acceptable in model development, cannot be considered unbiased.

We have removed this phrase following the reviewer’s suggestion.

Line 226, page 9: The term “iterative selection” is unclear. The authors should clarify what this process entailed.

To avoid confusion, we now removed the phrase “iterative selection” as the sentences preceding this phrase in that paragraph already reflected the selection procedure that we unbiasedly identified differentially expressed genes between resistant and sensitive tumors and calculated their AUC, and we then added back the genes from the pathway analyses and the *a priori* breast cancer genes.

Line 459, page 19: The authors should specify the exact sample size of the cohort used to develop the score (i.e., patients with RNA-seq data). Based on the current study, this appears to be $n = 20$, but it is unclear whether additional samples were included.

The reviewer is correct that 20 patients have RNA-seq data. The sample size of 20 is clearly indicated in this sentence and in the flow diagram in Supplemental Figure 1: “Baseline (BL) RNA-seq was performed for 20 patients (13 Sensitive and 7 Resistant) with sufficient tumor material (Supplemental Fig. 1)”.

Line 568, page 24: The statement “This provided validation...” should be tempered. The results supporting the gene signature are fragile (very small sample sizes for the development and validation cohorts, different outcomes for the development and the validation of the score) and should be presented with appropriate caution. As mentioned by the authors in the rebuttal letter “future large studies are needed to confirm and generalize these findings.”

We agree that future larger studies are needed and now add a sentence after the statement to acknowledge the limited sample size and call for future validation studies.

Reviewer #5 (Remarks on code availability):

The code is provided, but cannot be run without the source file, which is understandable given that it contains patient data. To enhance clarity, additional annotations would be helpful, in particular for the procedure of correction of AUC for optimism (see previous comment).

Response: We have now annotated the R function for AUC correction and also other functions in the codes.